# The formation and mitigation of nitrate pollution: Comparison between urban and suburban environments

Suxia Yang[1,2], Bin Yuan[1,2*], Yuwen Peng[1,2], Shan Huang[1,2], Wei Chen[3], Weiwei Hu[3], Chenglei Pei[3,4,5,6], Jun Zhou[1,2], David D. Parrish[1], Wenjie Wang[7], Xianjun He[1,2], Chunlei Cheng[2,8], Xiaobing Li[1,2], Xiaoyun Yang[1,2], Yu Song[7], Haichao Wang[9], Jipeng Qi[1,2], Baolin Wang[10], Chen Wang[10], Chaomin Wang[1,2], Zelong Wang[1,2], Tiange Li[1,2], E Zheng[1,2], Sihang Wang[1,2], Caihong Wu[1,2], Mingfu Cai[1,2], Chenshuo Ye[7], Wei Song[3], Peng Cheng[8], Duohong Chen[6], Xinming Wang[3], Zhanyi Zhang[1,2], Xuemei Wang[1,2], Junyu Zheng[1,2], Min Shao[1,2*]

[1]Institute for Environmental and Climate Research, Jinan University, Guangzhou 511443, China

[2]Guangdong-Hongkong-Macau Joint Laboratory of Collaborative Innovation for Environmental Quality, Jinan University, Guangzhou 511443, China

[3]State Key Laboratory of Organic Geochemistry and Guangdong Key Laboratory of Environmental Protection and Resources Utilization, Guangzhou Institute of Geochemistry, Chinese Academy of Sciences, Guangzhou 510640, China

[4]CAS Center for Excellence in Deep Earth Science, Guangzhou, 510640, China

[5]University of Chinese Academy of Sciences, Beijing 100049, China

[6]Guangzhou Ecological and Environmental Monitoring Center of Guangdong Province, Guangzhou 510060, China

[7]State Joint Key Laboratory of Environmental Simulation and Pollution Control, College of Environmental Sciences and Engineering, Peking University, Beijing 100871, China

[8]Institute of Mass Spectrometry and Atmospheric Environment, Guangdong Provincial Engineering Research Center for on-line Source Apportionment System of Air Pollution, Jinan University, Guangzhou 510632, China

[9]School of Atmospheric Sciences, Sun Yat-Sen University, Guangzhou 510275, China

[10]School of Environmental Science and Engineering, Qilu University of Technology, Jinan 250353, China

*Correspondence to*: Bin Yuan (byuan@jnu.edu.cn) and Min Shao (mshao@pku.edu.cn)

**Abstract.** Ambient nitrate has been of increasing concern in $PM_{2.5}$, while there are still large uncertainties in quantifying the formation of nitrate aerosol. The formation pathways of nitrate aerosol at an urban site and a suburban site in the Pearl River Delta (PRD) are investigated using an observation-constrained box model. Throughout the campaigns, aerosol pollution episodes were constantly accompanied with the increase of nitrate concentrations and fractions at both urban and suburban sites. The simulations demonstrate that chemical reactions in the daytime and at night both contributed significantly to formation of nitrate in the boundary layer at the two sites. However, nighttime reactions predominately occurred aloft in the residual layer at the urban site and downward transport from the residual layer in the morning is an important source (53%) for surface nitrate at the urban site, whereas similar amounts of nitrate were produced in the nocturnal boundary layer and residual layer at the suburban site, which results in little downward transport of nitrate from the residual layer to the ground at the suburban site. We show that nitrate formation was in the volatile organic compounds (VOCs)-limited regime at the urban site, and in the transition regime at the suburban site, identical to the response of ozone at both sites. The reduction of VOCs emissions can be an efficient approach to mitigate nitrate in both urban and suburban areas through influencing hydroxyl radical (OH) and $N_2O_5$ production, which will also be beneficial for the synergistic control of regional ozone pollution. The results highlight that the relative importance of nitrate formation pathways and ozone can be site-specific, and the quantitative understanding of various pathways of nitrate formation will provide insights for developing nitrate and ozone mitigation strategies.

**Keywords**: nitrate, ozone, volatile organic compounds, $N_2O_5$, formation pathways, urban and suburban sites

## 1 Introduction

Particulate nitrate is a substantial chemical component of fine particles, which plays a significant role in the acid deposition, visibility reduction, hygroscopic properties, and radiative forcing (Li et al., 1993;Watson, 2002;Pathak et al., 2009;Xu and Penner, 2012;Zhang et al., 2017;Liu et al., 2020). Due to the larger emission reduction of $SO_2$ than $NO_x$ and little change of $NH_3$ since the implementation of the clean air actions in China (Guo et al., 2018;Liu et al., 2019a;Zhai et al., 2021), a considerable increase in the nitrate fractions in aerosols has been observed in haze periods in the northern China Plain (Wen et al., 2018;Li et al., 2018;Lu et al., 2013;Fu et al., 2020), southern China (Pathak et al., 2009;Pathak et al., 2011) and eastern China (Griffith et al., 2015;Tao et al., 2018;Yun et al., 2018b;Li et al., 2018), which indicates the growing significance of nitrate in the formation of haze events. In addition, the photolysis of particulate nitrate can increase the production of sulfate and nitrous acid (HONO), implying the importance of nitrate in the synergetic enhancement of the atmospheric oxidizing capability in haze events (Gen et al., 2019;Zhang et al., 2020;Ye et al., 2016;Ye et al., 2017) , although the photolysis of particulate nitrate to produce HONO still remains highly uncertain (Romer et al., 2018). Hence, identifying and understanding the driving factors of nitrate formation are essential to establishment of optimized mitigation policies for fine particles.

Particulate inorganic nitrate is primarily produced through two processes: the photochemical reaction of hydroxyl radical (OH) and $NO_2$ during daytime (R1) and the heterogeneous uptake of $N_2O_5$ (R2–R5) during nighttime. The gaseous nitric acid ($HNO_3$) is produced by the reaction of OH and $NO_2$, and then reacts with ammonia ($NH_3$) to form particulate nitrate (Stelson and Seinfeld, 1982). The partitioning process of $HNO_3$ between the gas and particle phase is regulated by ambient temperature ($T$), relative humidity (RH) (Mozurkewich, 1993), aerosol pH and the abundance of $NH_3$ (R7) (Xue et al., 2014a;Yun et al., 2018b;Franchin et al., 2018). The pH value within a certain range plays an important role in the gas-particle partitioning of nitrate, which significantly impacts the nitrate formation (Guo et al., 2018;Lawal et al., 2018;Nenes et al., 2020).

$$OH + NO_2 \xrightarrow{k_1} HNO_3(g) \qquad\qquad R1$$

$NO_2 + O_3 \overset{k_3}{\rightarrow} NO_3 + O_2$                                   R2
$NO_3 + NO_2 \overset{k_4}{\leftrightarrow} N_2O_5$                                   R3
$NO_3 + VOCs \rightarrow products$                                                   R4
$N_2O_5 + (H_2O \text{ or } Cl^-) \overset{k_5}{\rightarrow} \varphi ClNO_2 + (2-\varphi)NO_3^-(p)$   R5
$k_5 = \frac{\omega 1 * \gamma * Sa}{4}$                                             (Eq.1)
$ClNO_2 \overset{k_6}{\rightarrow} Cl \cdot + NO_2$                                   R6
$k_6 = J_{ClNO_2}$                                                                   (Eq.2)
$HNO_3(g) + NH_3(g) \overset{k_7}{\leftrightarrow} NO_3^-(p) + NH_4^+(p)$             R7
The heterogeneous uptake reaction of $N_2O_5$ can occur on the surface of water or
chlorine-containing particle (R5); and the reaction constant ($k_5$) is described by Eq.1,
where $\varphi$ is the production yield of $ClNO_2$ in R5, $\omega_1$ is the average molecular speed of
$N_2O_5$ (m s$^{-1}$), $\gamma$ is the uptake coefficient of $N_2O_5$ and $S_a$ (m$^2$ m$^{-3}$) is the aerosol surface
area concentration in Eq.1. The nitryl chloride ($ClNO_2$) produced by the heterogeneous
uptake reaction of $N_2O_5$ at night would photolysis in the next morning, which would
produce chlorine atom and $NO_2$ (R6). Here the reaction rate $k_6$ was denoted as the
photolysis rate of $ClNO_2$ ($J_{ClNO_2}$). The heterogeneous uptake reaction of $N_2O_5$ is
affected by the uptake coefficient ($\gamma$) and the production yield of $ClNO_2$ ($\varphi$), which
cannot be directly measured and are significantly impacted by the aerosol components
and ambient RH (Bertram and Thornton, 2009;Bian et al., 2017;McDuffie et al.,
2018a;McDuffie et al., 2018b). Thus, the nocturnal contribution to nitrate formation
still has great uncertainty.
With the radiative cooling in the afternoon, the mixed layer decoupled into a
steady, near surface nocturnal boundary layer (NBL) and a residual layer (RL), which
is a neutral layer and formed aloft during the turbulence attenuation process (Prabhakar
et al., 2017). The heterogeneous uptake of $N_2O_5$ in the nocturnal boundary layer is
greatly disturbed in the presence of fresh NO emissions, which titrate the $NO_3$ radical
within the stagnant boundary layer (Geyer and Stutz, 2004;Li et al., 2020;Chen et al.,
2020). However, aircraft observations in California and Utah in the US have revealed
that active uptake of $N_2O_5$ in the residual layer contributed a major portion of the near-
surface nitrate accumulation during the morning transport from aloft (Brown et al.,
2006;Chow et al., 2006;Prabhakar et al., 2017;McDuffie et al., 2019;Womack et al.,

2019). Similarly, ground- and tower-based field observations also pointed out the important contribution of this pathway to the rapid increase of near-surface nitrate concentrations in Beijing, China (Wang et al., 2018a;Chen et al., 2020). However, under different atmospheric conditions, the relative importance of nitrate production varies significantly within the residual layer (McDuffie et al., 2019;Tang et al., 2021), giving widely varying relative contributions of the major chemical pathways to nitrate pollution among different sites (Wang et al., 2018a;Womack et al., 2019;Chen et al., 2020;Lin et al., 2020). A comprehensive understanding of the nitrate production in the residual layer is required to quantify the contributions of different formation pathways to nitrate pollution.

The nitrate production from the reaction of OH and $NO_2$ pathway during daytime is well-understood, and the control of $NO_x$ emission is commonly considered as an effective strategy to reduce ambient nitrate. However, several studies reported that the efficiency of $NO_x$ reduction in nitrate control is limited, and it may enhance nitrate production under some conditions (Womack et al., 2019;Dong et al., 2014;Hou et al., 2019). The study by Womack et al. (2019) showed that both nitrate and ozone were VOCs-limited in Salt Lake City, suggesting that VOCs control would effectively reduce nitrate. Similarly, modeling studies also found that the nitrate formation was more sensitive to the change in VOCs concentrations over the northern and eastern China (Dong et al., 2014;Lu et al., 2019;Fu et al., 2020). The sensitivity of nitrate production to both $NO_x$ and VOCs in different regions should be comparatively evaluated, which could provide helpful implications in formulating effective control strategies for the mitigation of aerosol pollution.

In recent years, the nitrate formation in haze episodes has been studied in northern China (Liu et al., 2015;Wang et al., 2017a;Wen et al., 2018;Fu et al., 2020;Chen et al., 2020), eastern China (Tao et al., 2016;Lin et al., 2020) and southern China (Qin et al., 2017;Tao et al., 2018;Yun et al., 2018b;Su et al., 2020), and the important contribution of the heterogeneous uptake of $N_2O_5$ in the nighttime has been discussed (Wang et al., 2017b;Yun et al., 2018b;Yun et al., 2018a;Chen et al., 2020). However, these ground-based observations rarely considered the potential contributions of reactive uptake of $N_2O_5$ aloft in the residual layer, which could be an important source of near-surface nitrate concentrations. In addition, few studies have comprehensively evaluated the relative influence of $NO_x$ and VOCs reductions on nitrate production in the urban and suburban areas (Hou et al., 2019).

In this study, we present the results from the ground- and tower-based measurements in both urban and suburban areas in southern China. An observation-constrained box model was used to simulate the production rates of nitrate from different formation pathways, and to compare the effects of reducing $NO_x$ and VOCs emissions in both urban and suburban areas. This work provides new insights into the synergetic mitigation of particle and ozone pollution, which can guide development of the most effective nitrate control strategies.

## 2 Method and data

### 2.1 Field observation

The ground-based field measurements were conducted at both an urban site in Guangzhou and a suburban site in Heshan. The tower-based measurements were conducted at an urban site in Guangzhou. The ground-based study in Guangzhou was carried out from late September to mid-November in 2018 at the Institute of Geochemistry (GIG), Chinese Academy of Sciences (23.1°N, 113.2°E), which is a typical urban site surrounded by a residential area and traffic avenues (Fig. 1). The instruments were deployed on the top of the 25-m building at GIG site. The ground-based measurement at the suburban site was performed from late September to mid-November in 2019 at the supersite of Heshan county (22.7°N, 112.9°E), which is approximately 50 km southwest to Foshan and 80 km southwest to Guangzhou, and is frequently influenced by anthropogenic emissions from upwind Guangzhou-Foshan mega-city areas. The tower-based measurements were conducted simultaneously at the ground and 448 m on the Canton Tower from late September to mid-November in 2018 concurrent with the measurements at the GIG site, which are approximately 5.7 km apart each other (Fig. 1).

The chemical components of $PM_1$, trace gases, and non-methane hydrocarbons (NMHC), and particle BC content and particle size distribution were both measured at the GIG and Heshan sites, whereas only trace gases ($NO_x$ and $O_3$) and meteorological parameters were measured at the Canton Tower site. The non-refractory chemical compositions of $PM_1$ (NR-$PM_1$), including organics (Org), sulfate ($SO_4^{2-}$), nitrate ($NO_3^-$), ammonium ($NH_4^+$), and chloride ($Cl^-$) were measured using a high-resolution time-of-flight aerosol mass spectrometer (HR-ToF-AMS, Aerodyne Research Inc., US) (Hu et al., 2016;Chen et al., 2021). Black carbon (BC) was measured using an

aethalometer (AE33, Magee Scientific Co., US). Particle number size distribution was
measured using a scanning mobility particle sizer with an aerodynamic diameter
ranging from 10 to 650 nm (SMPS, TSI, US) and aerosol particle sizer ranging from
500 nm to 20 μm (APS, TSI, US). Details on the limit of detection and accuracy of the
instruments are presented in Table S1~ Table S3.
$HNO_3$, $N_2O_5$, and $ClNO_2$ were measured using iodide-time-of-flight chemical
ionization mass spectrometry (Iodide-TOF-CIMS, Aerodyne Research Inc., US) (Wang
et al., 2020b;Ye et al., 2021). The non-methane hydrocarbons (NMHC) were measured
using online GC-MS-FID (Wuhan Tianhong Co., Ltd, China) (Yuan et al., 2012) (Table
S4). The concentrations of oxygenated VOCs (OVOCs), including formaldehyde
(HCHO) and acetaldehyde ($CH_3CHO$), the sum of methyl vinyl ketone (MVK) and
methacrolein (MACR) were measured via a high-resolution proton transfer reaction
time-of-flight mass spectrometry (PTR-ToF-MS, Ionicon Analytik, Austria) (Wang et
al., 2020a;Wu et al., 2020). HONO was detected using a long path absorption
photometer (LOPAP) at the GIG site (Yu et al., 2021), and was measured by the gas
and aerosol collector (GAC) instrument  at the Heshan site (Dong et al., 2012;Yang et
al., 2014). $NH_3$ was also measured by two sets of instruments: a cavity ring-down
spectroscopy (CRDS, Picarro, US) was used at the GIG site and the GAC instrument
was used at the Heshan site (von Bobrutzki et al., 2010).
In addition, trace gases ($O_3$ (49i), $NO_x$ (42i), CO (48i) and $SO_2$ (43i)) (Thermo
Scientific, US) and meteorological parameters (i.e., wind speed (WS), wind direction
(WD), temperature (*T*), relative humidity (RH) and pressure (*P*)) (Vantage Pro 2, Davis
Instruments Co. , US) were simultaneously measured during these campaigns. The
photolysis frequencies of $O_3$, $NO_2$, HCHO, and HONO (PFS-100, Focused Photonics
Inc., China) were also measured during the campaigns. Considering the integrity and
temporal coverage of the measurements, we mainly focus on the investigated periods
from October 7 to 29, 2018, at the GIG site and from October 16 to November 16, 2019,
at the Heshan site.

## 2.2 Box Model description

A zero-dimensional observation-based box model (F0AM) (Wolfe et al., 2016)
was used to simulate the production of nitrate in this study. The F0AM box model uses
a subset of the Master Chemical Mechanism (MCM) v3.3.1 (Saunders et al.,
2003;Jenkin et al., 2003;Bloss et al., 2005), which explicitly describe chemical
reactions of VOCs, $RO_x$ radicals (including OH, $HO_2$ and $RO_2$), ozone and nitrate, and
was widely used in laboratory and theoretical researches (Edwards et al.,
2017;Anderson et al., 2017;D'Ambro et al., 2017;Womack et al., 2019).
In this study, the box model was constrained by observations of NMHCs, HCHO,
$CH_3CHO$, NO, CO, $CH_4$, HONO, and meteorological parameters (i.e., photolysis rates,
RH, $T$ and $P$) measured at the GIG and Heshan sites. To investigate the convection of
nitrate between the residual layer and the surface, the box model was split into two
boxes at night (from 17:00 to 6:00 of the following morning) to separately represent the
nocturnal boundary layer and the residual layer, respectively (Womack et al. (2019)
(Fig. S1).
The simulation of the residual layer at the GIG site was constrained by the
observation data from 488 m at the Canton Tower, while the simulation of the residual
layer at the Heshan site was freely evolved from sunset time using the ground
observation data of Heshan. The detailed model settings are described in Text S1, and
the agreement between the observation data and simulations at the GIG and Canton
Tower sites supports the use of similar simulation of the residual layer at the Heshan
site. The model was operated in a time-dependent mode with a 5-min resolution. It was
run for a 72-hour spin-up time to build steady-state concentrations for secondary
pollutants that were not constrained during simulation. To prevent the build-up of long-
lived species to unreasonable levels, an additional physical dilution process was applied
in the model(Lu et al., 2017;Decker et al., 2019;Novak and Bertram, 2020;Liu et al.,
2021;Yun et al., 2018b). To achieve agreement with the observation, a life time of 24
h and 8 h were used at the GIG and Heshan site, respectively. The sensitivity tests with
different dilution constant at the GIG and Heshan site were shown in Fig.S2 and Fig.S3,
respectively. The background concentrations for ozone and $CH_4$ were set as 30 ppb and
1.8 ppm, respectively (Wang et al., 2011).
The nocturnal production of nitrate from $N_2O_5$ hydrolysis and the subsequent
reactions (R5 and R6) are added to the box model. $\gamma$ and $\varphi$ are calculated using the
observation-based empirical parameterization method from Yu et al. (2020), where the
impacts of nitrate, chloride, and aerosol liquid water content (ALWC) were evaluated
to better represent the observed $\gamma$. The average values of $\gamma$ were 0.018±0.01 and
0.019±0.01 at the GIG and Heshan sites, respectively, which were comparable with the
observed mean data of γ (0.020±0.019) at the Heshan site in 2017. The φ used in this
study were 0.18±0.15 and 0.20±0.23 at the GIG and Heshan sites, which were slightly
lower than the observed mean data of φ at the Heshan site (0.31 ± 0.27) in 2017 (Yu et
al., 2020). The chemical compositions of fine particle were not measured at the Canton
Tower site, thus values of γ and φ in the residual layer were assigned equal to those of
the nocturnal boundary layer. The γ and φ exhibited complicated nonlinear dependence
on aerosol composition, aerosol liquid water and RH (Bertram and Thornton,
2009;McDuffie et al., 2019;Yu et al., 2020), such that γ and φ has positive and negative
dependence with RH, respectively. There was higher RH, and lower chloride at the 488
m site, compared to the ground site of Canton Tower. The nitrate concentration was
comparable at the 488 m site to the ground site in the study of Zhou et al. (2020).
Combined with the higher RH and lower $PM_{2.5}$ concentrations in the residual layer in
this study (as shown in Fig.S4), we inferred the negative deviations for γ and positive
deviations for φ in the residual layer. The dry aerosol surface area concentration ($S_a$)
was calculated from the particle number size distribution and calibrated to the actual
atmospheric $S_a$ using the RH-dependent hygroscopic growth factor (*f*(RH)). The f(RH)
was estimated from the aerosol composition measured by AMS and the aerosol liquid
water content, which included the inorganic-associated and organic-associated water.
The sum of inorganic-associated water estimated from ISORROPIA thermodynamic
model and organic-associated water estimated from the dry organic aerosol mass, was
used to calculate the growth of wet matter contributions, as described in the study of
McDuffie et al. (2018a). $J_{ClNO_2}$ was scaled from measured $NO_2$ photolysis frequencies
divided by a factor of 30 (Riedel et al., 2014).
The equilibrium coefficient between $HNO_3$ and particulate nitrate is incorporated
into the box model as a pseudo-first-order reaction (Eq.3 and 4) through the equilibrium
absorption partitioning theory (Jacob, 2000;Yuan et al., 2016):
$HNO_3 \text{ (g)} \underset{k_{8b}}{\overset{k_{8f}}{\Longleftrightarrow}} NO_3^- \text{ (p)}$      R8
$k_{8f} = \left(\frac{R_a}{D_g} + \frac{4}{\omega * \alpha}\right)^{-1} * S_a$      (Eq.3)
$k_{8b} = \left(\frac{R_a}{D_g} + \frac{4}{\omega * \alpha}\right)^{-1} \frac{S_a}{K_{eq}}$      (Eq.4)
where $R_a$ is the radius of nitrate particles (m), $D_g$ is the gas-phase molecular diffusion
coefficient ($m^2 s^{-1}$), ω is the mean molecular speed of $HNO_3$ ($m s^{-1}$), α is the mass
accommodation coefficient of $HNO_3$, and $K_{eq}$ represents the equilibrium constant of
$HNO_3$ and nitrate. These coefficients are the same as those in the chemical aqueous-
phase radical mechanism (CAPRAM) (Ervens et al., 2003;Wen et al., 2015).

The empirical kinetic modeling approach (EKMA) is used here to identify the

sensitivity of ozone and nitrate to the variations of $NO_x$ and VOCs. The observed
diurnal average conditions are used as the input for the base simulation. Sensitivity tests
are conducted by increasing and decreasing initial anthropogenic VOCs (AVOCs) and
$NO_x$ concentrations by a ratio ranging from 0.1 to 2.0 with 20 equal-distance steps
without changing other parameters in the model (Tan et al., 2018;Lyu et al.,
2019;Womack et al., 2019). The maximum concentration of ozone and nitrate in each
scenario are plotted to generate the contour plots of the respective isopleths. Isoprene
was included in the simulation as biogenic VOC (BVOC). Reducing BVOCs such as
isoprene is impractical, so it is not scaled with AVOCs concentrations in the sensitivity
simulations on control of precursors.

Since the $N_2O_5$ is affected by the chemistry between ozone and VOCs,

constraining $N_2O_5$ concentrations with the change in $NO_x$ ratio arbitrarily during the
isopleth simulations is improper. Thus, we set the simulation of base case (S0) without
$N_2O_5$ constrained. To evaluate the results of the base case, we design another simulation
with $N_2O_5$ constrained (S1) and compare the two simulated nitrate with the observation
in Fig. S5. The model scenarios were described in Table S5 in detail. The base case
simulation (S0) was comparable to the observation. The simulated nitrate with $N_2O_5$
constrained (S1) during October 9 to 10, 2018 was observed to be much higher
compared to both the observations and base case simulation (S0) at the GIG site, which
suggest that high concentrations of ambient $N_2O_5$ measured during this short period
may not contribute significantly to nitrate formation (Fig. S6). Overall, the simulated
nitrate of base case without $N_2O_5$ constrained agreed well with the observation
suggesting the robustness of the model simulations.

Gaussian error propagation was used to evaluate the uncertainties about

measurement parameters and reaction rates in the model, as described in Lu et al. (2012).
The uncertainties of various measurement parameters (VOCs, trace gases,
meteorological parameters, etc.) ranged from 0 to 20%, and uncertainties of reaction
rates are in the order of ~20% (Lu et al., 2012).

## 3 Results and discussion

## 3.1 Overview of nitrate concentrations during the campaign

The temporal variations of mass concentrations of the major chemical components in $PM_1$ are shown in Fig. 2. The mean concentration of $PM_1$ was $41.7\pm23.1$ μg m$^{-3}$ at the GIG site during the investigated period, which was comparable with that at the Heshan site ($40.6\pm15.5$ μg m$^{-3}$). The aerosol composition differed between sites, with inorganic ions (sulfate, nitrate, and ammonia) higher and organic matter lower at the GIG site compared to the Heshan site.

Although the mass concentrations at the two sites were comparable, the mass fraction of nitrate in $PM_1$ at the GIG site increased from 10% to 33% as the mass concentration of $PM_1$ increased from 20 to 130 μg m$^{-3}$ (Fig. 3), while the fraction of nitrate increased from 10% to 20% at the Heshan site, suggesting that nitrate plays a more important role in the increase in $PM_1$ at the urban site than that at the suburban site. The significant increasing ratio of nitrate fraction from clean condition to polluted condition ($\sim 43\%$) was also revealed in the airborne observation in Utah Valley, US (Franchin et al., 2018). In addition, although the concentration of sulfate was higher than that of nitrate during most of the sampling periods, as $PM_1$ increased the mass concentration ratio of nitrate/sulfate increased from 0.5 to 2.0 at the GIG site and from 0.5 to 1.5 at the Heshan site. The higher ratios of nitrate/sulfate during the polluted periods implies that reducing nitrate may be essential for reducing the occurrence of PM pollution in southern China. The increasing contributions of nitrate to $PM_1$ in this study were similar with those observed in northern China during haze pollution (Yang et al., 2017;Fu et al., 2020;Wen et al., 2015;Liu et al., 2015), suggesting the significance of nitrate mitigation to further reduce mass concentrations of fine particles in China.

The diurnal patterns of mean nitrate, $NH_3$, $NO_2$ and $HNO_3$ concentrations observed at the GIG and Heshan sites are shown in Fig. 4. The highest nitrate concentration was observed in the morning at the GIG site and during nighttime at the Heshan site, suggesting differences in the processes that dominated the formation of nitrate at the two sites. At the GIG site, nitrate rapidly increased from 4:00 to 9:00, but the concentrations of $NH_3$ and $HNO_3$ increased slowly, which suggests the minor contribution of direct production of $HNO_3$ from the reaction of OH and $NO_2$. The increase of nitrate during this period might be associated with the downward transport from the residual layer to the ground. The diurnal variations in $O_3$ and $NO_x$ measured

at the GIG and Canton Tower sites are shown in Fig. 5. The ground-based observations
at the Canton Tower showed similar variation patterns of $O_3$ and $NO_x$ to the GIG site.
However, the average concentration of $O_3$ at 488 m of Canton Tower site was 2.4 times
higher than that at the GIG site during nighttime, and the lower nocturnal concentrations
of NO (nearly zero) at the 488 m site would enhance the production of $NO_3$ and $N_2O_5$
(Wang et al., 2018b;McDuffie et al., 2019). Therefore, heterogeneous uptake of $N_2O_5$
during nighttime may be active at 488 m at urban site, which will be further investigated
in Section 3.2. At the Heshan site, nitrate increased sharply in the early nighttime
(before midnight), which may be attributable to the shallow nocturnal boundary layer
or the enhanced nocturnal $N_2O_5$ heterogeneous uptake reactions. Subsequently, there
was a significant increase in nitrate from 7:00 to 9:00. The concentration of $NH_3$
showed variation pattern that was similar with that of nitrate and increased after 7:00,
while the concentrations of $HNO_3$ and $NO_2$ showed a decreasing trend from 7:00 to
9:00 at the Heshan site. The different growth characteristics of nitrate and the variation
patterns of precursors at the two sites may be related to different formation processes,
which will be discussed in detail later.

In this study, the wind speeds in the investigated periods at the GIG and Heshan

sites were generally below 2 m s$^{-1}$ (Table S6), which suggests that regional transport
may have limited contributions to the abundance of nitrate at the observation sites.
Therefore, the discussion of the chemical formation process of nitrate in this study
focuses on local production.

The molar ratios of $[NH_4^+]$ to the sum of $2\times[SO_4^{2-}]+[NO_3^-]$ are calculated (Fig. S7)

to determine whether there was enough $NH_4^+$ to neutralize nitrate. The molar ratios
were approximately 1.0 at both GIG and Heshan sites, suggesting both $NH_3$ and $HNO_3$
were crucial precursors for nitrate formation. Based on these discussions, we will
discuss the $NH_3$ effect on the nitrate partitioning firstly by thermodynamic
ISORROPIA II model. The nitrate chemical formation pathways, which is mainly
attributable to the production of $HNO_3$ and/or heterogeneous uptake of $N_2O_5$ combining
the box model, will be discussed in Sec. 3.2.

The ISORROPIA II model setting is described in Test S2 in detail. The

ISORROPIA II modeled results of nitrate, ammonium, $HNO_3$, and $NH_3$ at the GIG and
Heshan site were displayed in Fig.S8 ~ Fig.S9. The particle-phase nitrate and
ammonium at the GIG site showed a bit overestimation, while the gas-phase $HNO_3$,
and $NH_3$ showed overestimation at the Heshan site. Overall, the simulated components
showed good correlations with the observed concentrations at both sites. We use the
ISORROPIA II model results to evaluate the particle fraction of nitrate in the sum of
$HNO_3$+nitrate ($\varepsilon(NO_3^-)$) against aerosol pH. Aerosol pH, which depends on the aerosol
acidity and water content, is calculated by the following equation:
$pH = -\log_{10} \dfrac{1000\, H_{air}^+}{ALWC}$                   (Eq.5)
where $H_{air}^+$ ($\mu g\ m^{-3}$) is the hydronium concentration of the equilibrium particle and
ALWC ($\mu g\ m^{-3}$) is the aerosol water content from ISORROPIA II simulation.

The $\varepsilon(NO_3^-)$ against pH at the GIG and Heshan site are shown in Fig.6. The pH

data are colored by relative humidity and fit to an "s-curve" as in Guo et al. (2018). The
clustering of pH data mainly located between 1~ 3, and the $\varepsilon(NO_3^-)$ are sensitive to the
change of pH. To further evaluate the sensitivity of $NH_3$ and sulfate on this effect, the
input of total ammonium (NHx, ammonium + $NH_3$) and sulfate were reduced from 10%
to 90% relative to the ISORROPIA II base model, respectively, while keeping other
parameters constant. The response of simulated nitrate concentration and aerosol pH to
changes in NHx and $SO_4^{2-}$ are shown in Fig.7. The nitrate concentration decreased with
the reduction of NHx, and had little variation with the reduction of $SO_4^{2-}$ (Fig.7 (a ~ b))
at both sites. Along with the reduction of NHx, the pH values decreased significantly
(Fig.7 (c ~ d)), which caused the further decrease of $\varepsilon(NO_3^-)$. The pH values showed a
bit increase with the reduction of $SO_4^{2-}$, which may be caused by that there would be
more available ammonium neutralized the hydronium. It is consistent with the study of
Guo et al. (2018) and Nenes et al. (2020), suggesting the partitioning of nitrate was also
affected by the $NH_3$ in the pH values between 1~3. Thus, the control of $NH_3$ is effective
for the reduction of nitrate by affecting the partitioning process of nitrate at both GIG
and Heshan site in this study. The partitioning of nitrate increased with the reduction of
sulfate suggests the limited role of sulfate reduction on the mitigation of nitrate.
**3.2 Contributions of different pathways to nitrate formation**

To further investigate the chemical formation pathways of nitrate, which related

to the photochemical and heterogeneous reactions, we adopt the box model results to
simulate the contribution of different pathways to nitrate formation. The temporal
variations in simulated and observed nitrate concentrations at the GIG and Heshan sites
are presented in Fig. 8; simulated and observed nitrate showed similar concentrations
and variation patterns. The diurnal variation of simulated nitrate is compared with the
observation in Fig.S10. The diurnal simulated nitrate was comparable with the
observation at the GIG site, especially when considering the vertical transport from the
residual layer in the morning. Unlike the GIG site, the diurnal simulated nitrate
performed higher in the daytime, and little bit lower in the late nighttime, compared
with the observation. It may be related to the lack of quantitative transport in the box
model. The box model performance was evaluated using the mean bias (MB), index of
agreement (IOA), and correlation coefficient ($r$) (Table S7) (Liu et al., 2019b;Lyu et al.,
2017;Wang et al., 2019;Curci et al., 2015). The IOA was larger than 0.7 and r was larger
than 0.5 at both sites, indicating good agreements between simulated and observed
nitrate concentrations. The temporal variations in simulated $N_2O_5$ and $ClNO_2$
concentrations were higher than the observations at the Heshan site as shown in Fig. S6
(c, d), the simulated results at the GIG site from October 9 to 10 were significantly
lower than the observations (Fig.S6 (a, b)). The abnormally high observed
concentrations of $N_2O_5$ and $ClNO_2$ that lasted for short periods (10-30 minutes) at the
GIG site may be caused by transported air masses from upwind regions or vertical
transport without well-mixed with fresh urban NO emissions. Simulation of these near-
instantaneous processes transported to the site using a box model is difficult, as box
model is more suitable to simulate the well-mixed airmass with little transport effects.
However, the simulated nitrate concentrations without observed $N_2O_5$ constrained was
adequately comparable with the observations as shown in Fig. S5, implying the
influence of the instantaneously high concentrations of $N_2O_5$ on nitrate formation was
negligible at the GIG site.
Based on these simulation results, we calculated the daily-averaged contributions
of the two different reaction pathways to the nitrate concentration - the daytime
production from OH + $NO_2$ reaction and the nighttime production from $N_2O_5$ uptake
reaction in the nocturnal boundary layer and in the residual layer. The nitrate produced
in the residual layer is only gradually mixed to the surface as the boundary layer
develops during the following morning, while the nitrate contributed to the boundary
layer column concentration always included the $N_2O_5$ uptake in the residual layer during
the whole nighttime (Wang et al., 2018a;Womack et al., 2019). The calculation methods
to determine contribution to the boundary layer column concentrations and to ground-
level nitrate concentrations should be distinguished.
To calculate the contribution to the boundary layer column concentration, the
integral of the nitrate production rate from $N_2O_5$ uptake from both the nocturnal surface
layer and the residual layer directly contribute to nitrate column concentrations layer
during the whole nighttime, weighted as 0.4 and 0.6 based on their altitude fractions of
the two layers, respectively. This calculation for the contributions to column
concentration is the same as the methods presented by Wang et al. (2018a) and Womack
et al. (2019). However, to quantify the contribution of nitrate produced from the
residual layer to the ground nitrate concentration, one must account for the dynamic
exchange between the residual layer and the surface-based boundary layer that develops
during daytime. The integral time for this dynamic exchange was assumed from 6:00
to 10:00 in the morning. Detailed descriptions of the calculations are provided in Text
S3 in Supplementary Materials. The calculation about partitioning process from OH
and $NO_2$ reaction in the daytime was the same in the two methods mentioned above,
which was the partition part of the integral of the OH and $NO_2$ reaction during the
daytime.
The contributions of nitrate to the boundary layer column concentration (i.e.
average from ground to 1000 m) are shown in Fig. 9a. The contribution of nitrate
production rate from $N_2O_5$ uptake in the residual layer was 17.9 $\mu g\ m^{-3}\ day^{-1}$ at the GIG
site, which was much greater than the $N_2O_5$ uptake in the nocturnal boundary layer (0.4
$\mu g\ m^{-3}\ day^{-1}$). This may be caused by the fresh NO surface emissions, which titrate the
$NO_3$ radical and ozone in the nocturnal boundary layer, as the mean NO concentration
during the nighttime at the GIG site was 12.1 ppb. The contribution from nocturnal
nitrate production in the boundary layer was comparable with the contribution from OH
and $NO_2$ reaction (13.2 $\mu g\ m^{-3}\ day^{-1}$) during the daytime. In contrast to the GIG site,
the contribution of nitrate production rate from $N_2O_5$ uptake in the nocturnal boundary
layer (6.2 $\mu g\ m^{-3}\ day^{-1}$) was comparable with that in the residual layer (4.4 $\mu g\ m^{-3}\ day^{-1}$
$^{1}$) at the Heshan site. The similar nitrate concentration and production rate from $N_2O_5$
uptake between the nocturnal boundary layer and residual layer in Fig.S12 (c, d) was
due to smaller NO emissions at the Heshan site. The results demonstrate that nocturnal
nitrate production plays an important role in nitrate production in the boundary layer,
with nighttime contributions of 58% at the urban site and 35% at the suburban site.
The relative magnitudes of the contributions to the daily-averaged surface nitrate
differ somewhat from the contributions to the entire boundary layer. The contributions
from the three major pathways to surface nitrate concentrations at the two sites are
compared in Fig. 9b. At the GIG site the nitrate production rate from the OH and $NO_2$
reaction and downward transport from the residual layer were 13.2 $\mu g\ m^{-3}\ day^{-1}$ and
16.6 $\mu g\ m^{-3}\ day^{-1}$, contributing 43% and 53% of ground-level nitrate concentrations,
with a minor contribution (1.1 $\mu g\ m^{-3}\ day^{-1}$) from the production of $N_2O_5$ uptake in the
nocturnal boundary layer. This is similar with the results in Fig.9a, implying a large
nitrate contribution from $N_2O_5$ uptake in the residual layer, but not in the nocturnal
boundary layer at the urban site.
However, at the suburban Heshan site (Fig.9b), downward transport from the
residual layer made no contribution to the surface nitrate concentration, which was
smaller than the contribution of nitrate from the residual layer in Fig. 9a. This is due to
the similar nitrate production rate from $N_2O_5$ uptake between the nocturnal boundary
layer and residual layer (see Fig. S12), inducing negligible convection between the two
layers as the result of small concentration gradient (Brown et al., 2003;Baasandorj et
al., 2017;Prabhakar et al., 2017). The nitrate production rate from OH and $NO_2$ reaction
(19.9 $\mu g\ m^{-3}\ day^{-1}$) and nocturnal $N_2O_5$ uptake (15.6 $\mu g\ m^{-3}\ day^{-1}$) were the major nitrate
formation pathways, which contributed 56% and 44% to the surface total nitrate
production, respectively. Therefore, the importance of residual layer contribution to the
surface nitrate can vary significantly and should be comprehensively evaluated in
different environments. In addition, the nitrate contributions to the surface
concentrations and boundary layer column concentrations can also be different in
different regions, which should be clarified and distinguished in future studies.
In summary, the $N_2O_5$ uptake reaction was active in the residual layer both at urban
and suburban sites, the downward transport from the residual layer was a significant
contributor to surface nitrate at the urban site, but not at the suburban site. This is
attributable to the titration of the $NO_3$ radical and ozone by fresh NO emissions during
the stagnant boundary layer at the urban site, resulting in the large difference of nitrate
production between the residual and nocturnal boundary layers. In contrast, at the
suburban site, lower NO emissions favored $NO_3$ production and heterogeneous uptake
of $N_2O_5$ both in the nocturnal boundary layer and the residual layer. The horizontal
transport in the residual layer from nocturnal jets may contribute to the different nitrate
production at urban and suburban sites, which has been discussed in the research of
Chow et al. (2006) and Brown et al. (2006). Due to the limitation of box model, this
issue could be studied by the chemistry transport model in further research.

## 3.3 Control of NOx and VOCs as mitigation strategies of nitrate

Overall, the contributions of nitrate from the three major pathways, all involving $NO_x$ and ozone, suggest that nitrate formation depends not only on the reactions of $NO_x$ but also is closely associated with the VOCs-$NO_x$-$O_3$ chemistry. Therefore, the influence of both $NO_x$ and VOCs reduction on nitrate production should be considered in formulating policies to control aerosol pollution.

In this study, we adopted the widely used EKMA approach, generally used for ozone sensitivity analysis (Edwards et al., 2014;Mazzuca et al., 2016;Xue et al., 2014b;Wang et al., 2015) to investigate the response of nitrate formation in changing emissions of VOCs and $NO_x$. The dependence of simulated nitrate concentrations with changing of VOCs and $NO_x$ concentration allow to construct isopleths of nitrate and ozone production at the GIG and Heshan sites, as displayed in Fig. 10. The production of nitrate and ozone were in the VOCs-limited regime at the GIG site, and in the transition regime at the Heshan site, where nitrate and ozone are sensitive to both VOCs and $NO_x$ reduction. As shown in Fig. 11, the reduction of $NO_x$ emissions from $0 \sim 70\%$ would increase nitrate and ozone concentrations at the GIG site, but decrease those concentrations at the Heshan site. The decrease in VOCs concentrations would decrease nitrate and ozone concentrations at both sites. These results suggest that control of VOCs emissions will efficiently reduce nitrate and ozone production in both urban and suburban areas, but control of $NO_x$ emissions will give different responses between urban and suburban area for both ozone and nitrate. Fig. 11 show that the nitrate sensitivity to the reduction of VOCs and $NO_x$ emissions was identical to the response of ozone at both sites. These results demonstrate the possibility of synergetic control for nitrate and ozone at both urban and suburban sites through VOCs control.

The accuracy of the isopleth plots in Fig. 10 depends on several variables and parameters included in the box model. Figs S13 ~14 show the results of simulation experiments on the dependence of the isopleths upon changing various parameterization for estimating HONO concentrations, $N_2O_5$ uptake coefficient, and $ClNO_2$ yields as described in Text S4. The sensitivity regime of nitrate and ozone did not change, although the peak concentrations of ozone and nitrate did change, which supports the reliability of the results discussed above.

As nitrate and ozone exhibit similar sensitivity to the reduction of $NO_x$ and VOCs, different VOCs/$NO_x$ ratios may point to different control strategies. In the cases of the

Heshan and GIG sites, the reduction of $NO_x$ can adequately control nitrate production
with a $VOCs/NO_x$ ratio of 1.8 at the Heshan site, while a contrary result can be found
at the GIG site (with a $VOCs/NO_x$ ratio of 0.8) with a less than 70% reduction of $NO_x$
emission. The simulated results at the GIG site agree well with those reported in the
urban areas of Shanghai in China (Dong et al., 2014) and the Salt Lake City and San
Joaquin Valley in the US (Betty and Christian, 2001;Womack et al., 2019), which all
emphasized the decrease of nitrate production with the reduction of VOCs emissions,
and the enhanced nitrate production with $NO_x$ reduction. The results at the Heshan site
were consistent with the simulations at the suburban site of northern China, where a
higher $VOCs/NO_x$ ratio was found (Wen et al., 2018;Lu et al., 2019). The synergetic
reduction of $NO_x$ and VOCs is necessary to effectively mitigate the nitrate production
in consideration of the different $VOCs/NO_x$ ratios in the urban and suburban areas.
The above discussions revealed that direct reduction of $NO_x$ may not lead to a
decrease in nitrate production. Meanwhile, the reduction of VOCs is effective to
mitigate nitrate production, though they were not the direct precursors of nitrate. To
illustrate these findings, the impacts of changing VOCs and $NO_x$ on the production rate
of the OH radical, and the rate of OH plus $NO_2$, and the $N_2O_5$ uptake reaction were
evaluated. During daytime nitrate production involves OH production and its
subsequent reaction with $NO_2$. As shown in Fig. 12, the $NO_x$-saturated condition at the
GIG site provided sufficient $NO_2$ to quench the OH radical during daytime. A less than
70%reduction of $NO_x$ will increase ozone production and thereby drive more
production of OH, leading to increase in the OH and $NO_2$ reaction rates. When $NO_x$ is
lower than 30% of the base case emissions, ozone production would decrease and lead
to the decrease of OH production and its reaction with $NO_2$, which in turn bring about
a decrease in nitrate production. In contrast, at the Heshan site, the base case $NO_x$
concentrations are lower, giving a production rate of OH that is already sensitive to
both $NO_x$ and VOCs reductions. The model results indicates that further emission
reductions in both $NO_x$ and VOCs will simultaneously mitigate the production of nitrate
and ozone.
During nighttime, the initial ozone concentration participated the nocturnal
chemistry increased/decreased with the reduction of $NO_x$ at the GIG/Heshan site. In
addition, the decrease in $NO_x$ will reduce the titration effect of NO on $NO_3$ radical and
ozone at the GIG site, which enhances production of $N_2O_5$ and promotes nitrate
production in both the nocturnal boundary layer and the residual layer (Fig.13).
However, at the Heshan site, the reduction of $NO_x$ cuts down the sources of $NO_2$ and
$NO_3$, decreasing the formation of $N_2O_5$ and thus its heterogeneous uptake to produce
nitrate. The reduction of VOCs decreases ozone formation during daytime, thus
attenuating the nocturnal formation of $NO_3$, $N_2O_5$ and nitrate at both the GIG and
Heshan sites.
In summary, nitrate and ozone show similar responses to the reduction of $NO_x$ and
VOCs for both daytime and nighttime chemical processes, as the result of the coupling
between the formation reactions of ozone and nitrate. The results of this study
emphasize the complex effects of reductions of $NO_x$ emissions on nitrate concentrations
in the urban and suburban areas. In addition, the reduction of VOCs emissions would
be effective in the concurrent mitigation of ozone and nitrate, suggesting that the
reduction of VOCs at present is an effective method for the synergistic control of ozone
and $PM_{2.5}$ at present. As there are limitations of box modeling, a comprehensive three-
dimensional model assessment is needed on a regional scale.

## 595 4 Conclusions

In this study, we use an observation-constrained box model to explore the nitrate
formation pathways and implications for nitrate mitigation strategies at urban and
suburban sites. At both sites, the mass fraction of nitrate in $PM_1$ increased as the
absolute $PM_1$ levels increased (from 10% to 33% at the urban site and from 10% to 20%
at the suburban site), suggesting the important role played by nitrate in increasing
particle concentrations in the PRD.
Both $HNO_3$ and $NH_3$ are important precursors for nitrate formation. Combined
with the ISORROPIA II thermodynamic model, the reduction of $NH_3$ is effective for
the nitrate reduction by affecting the partitioning process of nitrate at both GIG and
Heshan site. The box model simulations demonstrate that chemical reactions in the
daytime and at night both contributed significantly to formation of nitrate in the
boundary layer at the two sites, with nighttime contributions of 58% at the urban site
and 35% at the suburban site. However, nighttime reactions predominately occurred
aloft in the residual layer at the urban site and downward transport from the residual
layer in the morning are important source (53%) for surface nitrate at the urban site,
whereas similar amounts of nitrate were produced in the nocturnal boundary layer and
residual layer at the suburban site, which results in little downward transport of nitrate
from the residual layer to the ground at this region. The spatial differences of nocturnal
reactions and the different contributions from downward transport of the residual layer
to surface nitrate at urban and suburban sites were attributed to different fresh emissions
and concentration levels of $NO_x$ at the two sites during the night time, suggesting that
nitrate production under different $NO_x$ conditions should be explored to better
understand the its formation pathways.
The non-linear relationships between nitrate and $NO_x$, VOCs was developed to
investigate the nitrate mitigation strategies. The simulations demonstrated that the
formation processes of both nitrate and ozone were in the VOCs-limited region at the
urban site and in the transition region at the suburban site. The same sensitivity regimes
of nitrate and ozone at two sites was caused by the similar chemical processes that
account to produce nitrate and ozone. These results suggest that control of VOCs
emissions would effectively mitigate nitrate in both urban and suburban areas.
Overall, the formation processes of nitrate are systematically investigated in both
urban and suburban areas in this study, which provides the opportunity to identify
different influencing factors of nitrate production in different environments and offers
insights into the comprehensive mitigation of nitrate pollution in regional scale. $NO_x$
emission controls alone might not be an effective strategy for reducing the nitrate
production, while the reduction of VOCs emissions would take effect in the concurrent
mitigation of ozone and nitrate. Thus, an emission control policy focusing on VOCs
will be an effective means for the synergistic control of ozone and $PM_{2.5}$ at present. In
the long-term, multi-pollutant control should be implemented to achieve better control
strategies for ozone and $PM_{2.5}$. As the result of limitation for the 0-D box model, vertical
transport and horizontal transport cannot be considered explicitly in this study. Given
the limitations of the box model, three-dimensional models should be used to further
investigate the synergistic control of ozone and particles on the regional scale.

## Data availability

The observational data used in this study are available from corresponding authors
upon request (byuan@jnu.edu.cn)

## Author contributions

BY and MS designed the research. SXY, YWP, SH, WC, WWH, CLP, CMW,
ZLW, TGL, EZ, MFC, XBL, SHW, CHW, WWJ, CSY, WS and PC contributed to data
collection. SXY performed the data analysis, with contributions from JZ,DD. Parrish,

XJH, CCL, XYY, YS, HCW, DHC, XMW, ZYZ, JYZ and XMW. SXY and BY prepared the manuscript with contributions from the other authors. All the authors reviewed the manuscript.

## Competing interests

The authors declare that they have no known competing financial interests or personal relationships that could have appeared to influence the work reported in this paper.

## Acknowledgments

This work was supported by the National Key R&D Plan of China (grant No. 2019YFE0106300, 2018YFC0213904, 2016YFC0202206), the National Natural Science Foundation of China (grant No. 41877302), the National Natural Science Foundation of China (grant No. 41905111), Guangdong Natural Science Funds for Distinguished Young Scholar (grant No. 2018B030306037), Key-Area Research and Development Program of Guangdong Province (grant No. 2019B110206001), Guangdong Soft Science Research Program (2019B101001005), and Guangdong Innovative and Entrepreneurial Research Team Program (grant No. 2016ZT06N263). This work was also supported by Special Fund Project for Science and Technology Innovation Strategy of Guangdong Province (Grant No.2019B121205004).

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

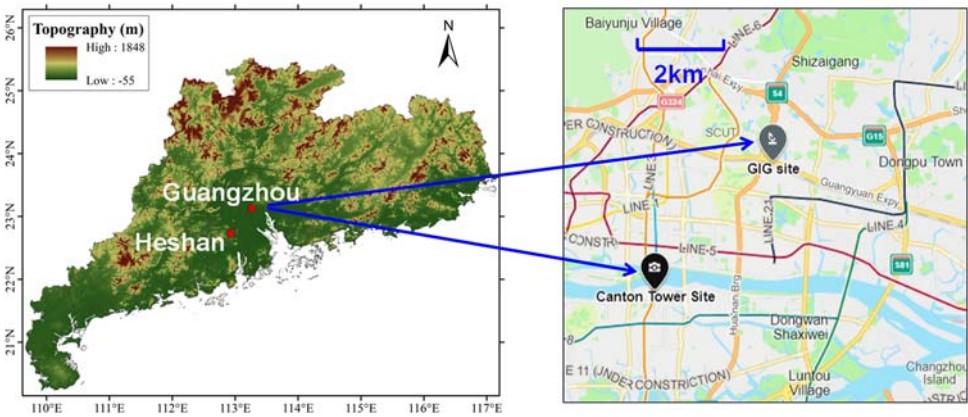


**Figure 1.** Sampling site at Guangzhou Institute of Geochemistry, Chinese Academy of
Sciences (GIG), Heshan and Canton Tower. Note that the map is extracted from
Microsoft Bing maps by the authors.


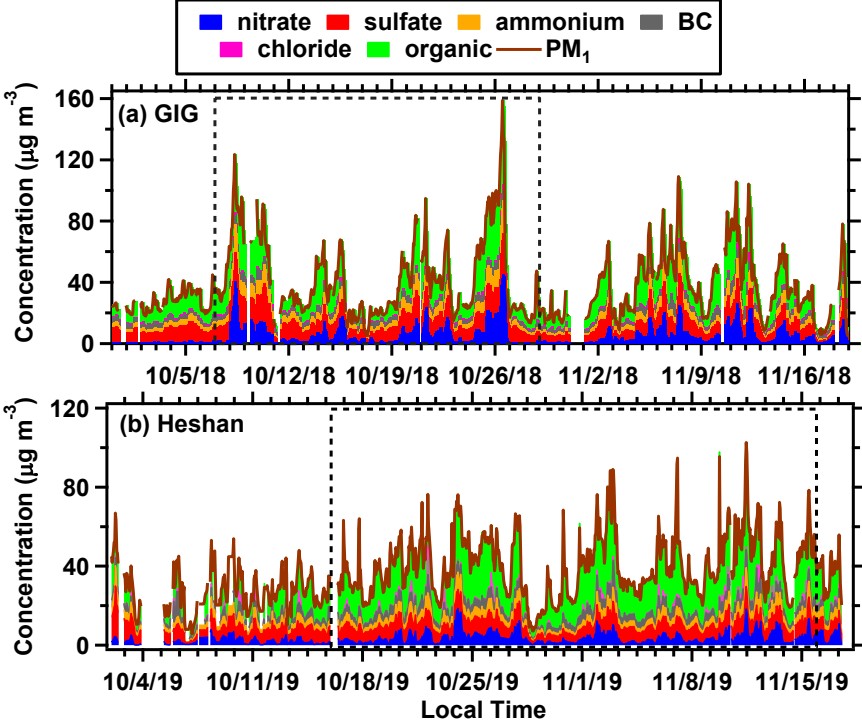


**Figure 2.** Temporal variations of the mass concentration of the major chemical components in PM$_1$ including nitrate (NO$_3^-$), sulfate (SO$_4^{2-}$), ammonium (NH$_4^+$), black carbon (BC), chloride (Cl$^-$) and organics at (a) GIG site and (b) Heshan site. The black dashed rectangle represents the investigated period which had complete set of data.







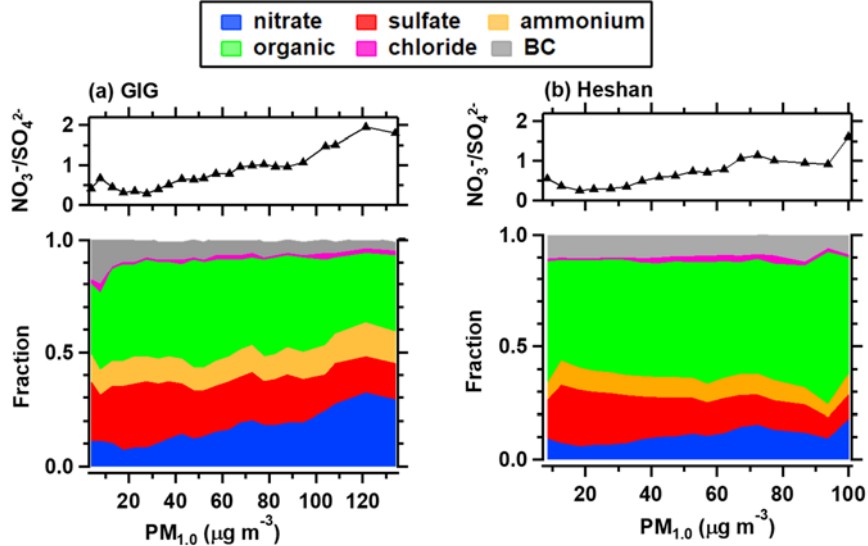


**Figure 3.** The mass concentration ratio of $NO_3^-/SO_4^{2-}$ (top) and fractions of major
chemical components (bottom) in $PM_1$ at (a) GIG site and (b) Heshan site.

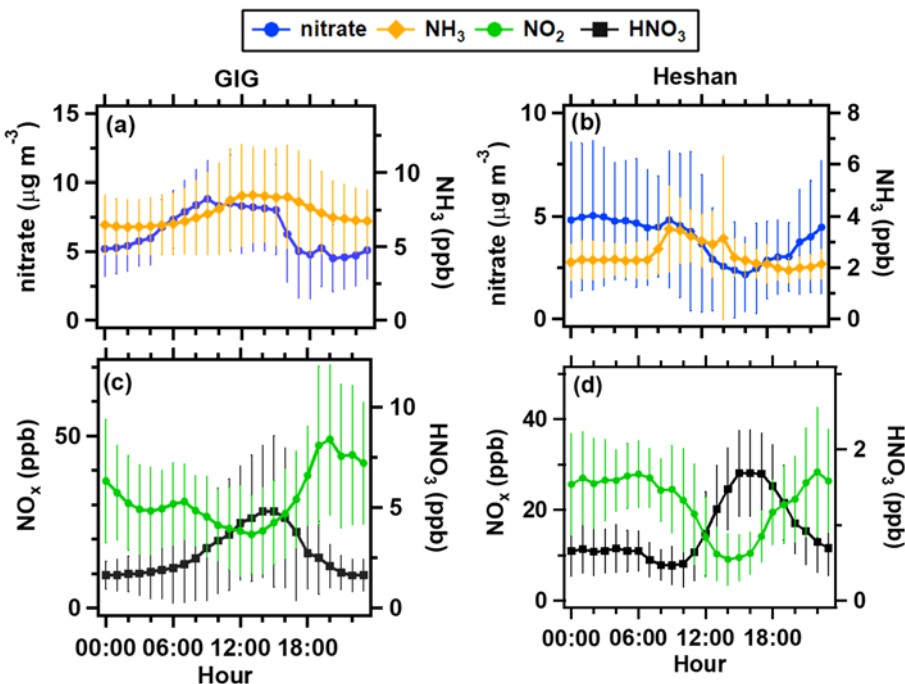


**Figure 4.** Diurnal variations of mean concentrations of nitrate and related pollution
species at (a) GIG site and (b) Heshan site. The error bars represent the standard
deviation of the means.


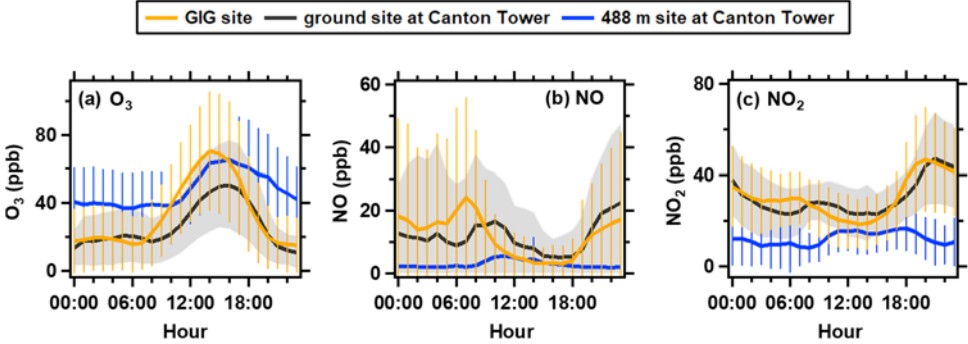


**Figure 5.** Diurnal variation of mean concentrations of (a) $O_3$, (b) NO, (c) $NO_2$ at GIG
(orange lines), and the ground and 488m sites of Canton Tower (black and blue lines,
respectively). The orange and blue error bars represent the standard deviations of the
mean concentrations at the GIG site and the 488m site of Canton Tower, the grey areas
show one standard deviation of the mean concentration at ground site of Canton Tower.


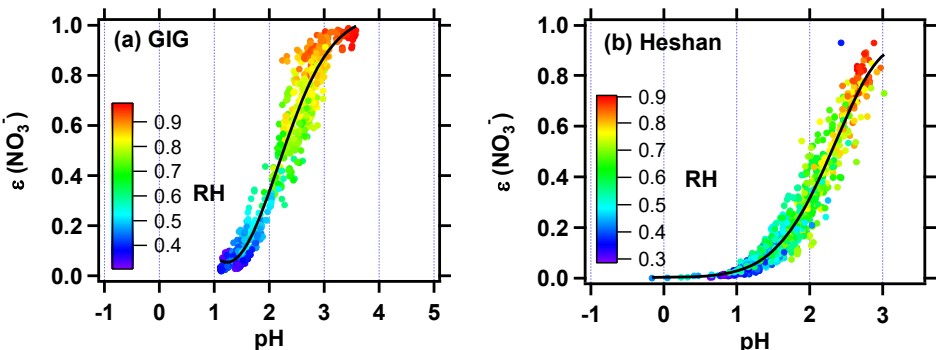


**Figure 6.** The particle fraction of nitrate in the sum of HNO₃+nitrate (ε(NO₃⁻)) against

aerosol pH. The pH data are colored by relative humidity and fit to an "s-curve" in

black line, as shown in Guo et al. (2018).

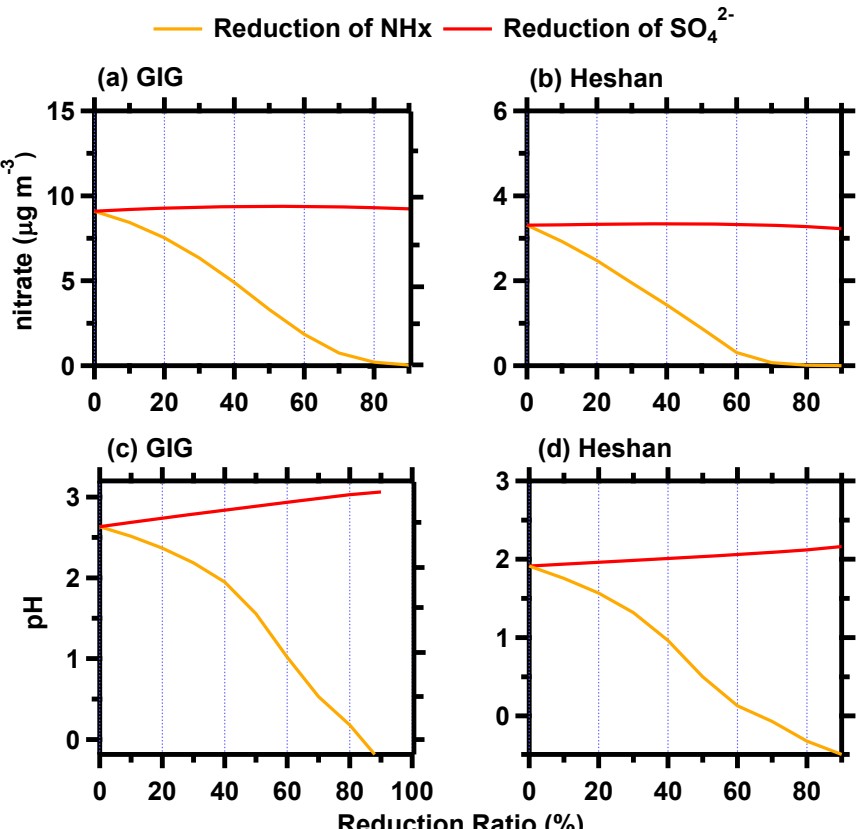

**Figure 7.** ISORROPIA-predicted average nitrate (a, b) and pH (c, d) as a function of

changes in NHx (ammonium + NH₃, orange line) and SO₄²⁻ (red line) at the GIG and

Heshan site during the study period.



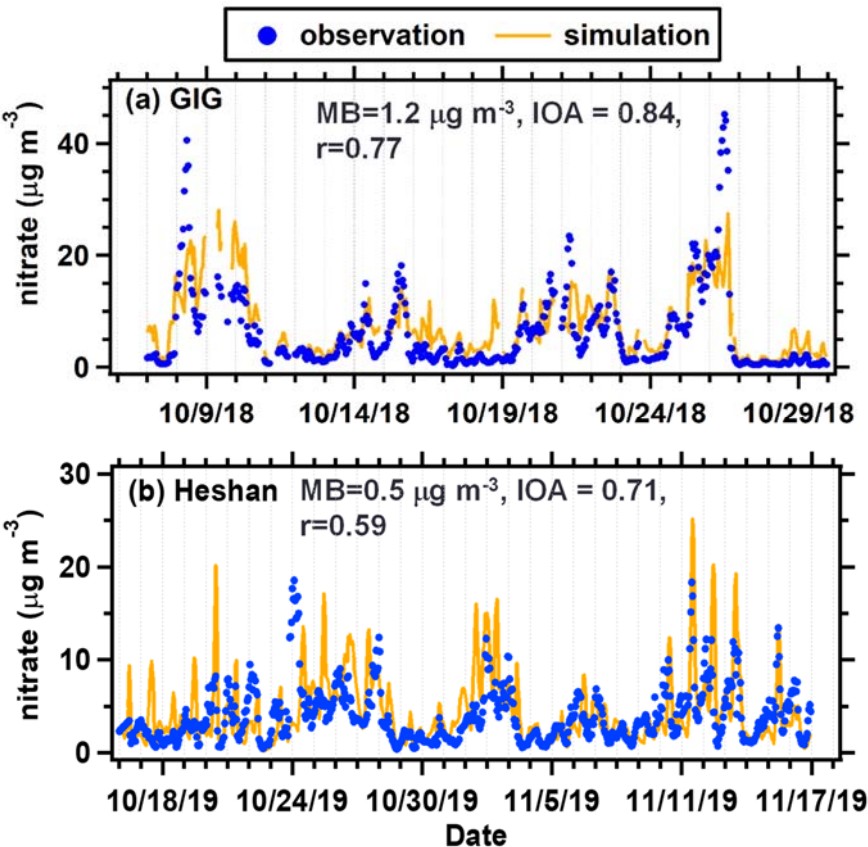


**Figure 8.** Comparison of the temporal box model simulated and observed nitrate at the
(a) GIG site and (b) Heshan site.

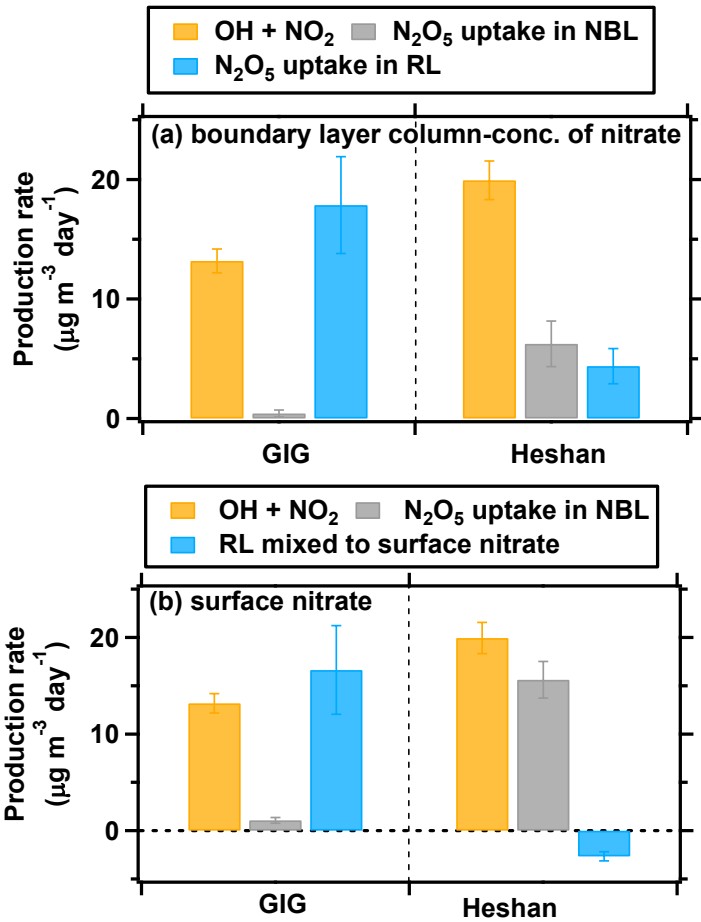


**Figure 9.** The daily-averaged contribution (a) to boundary layer column concentration
and (b) to surface nitrate from three pathways (OH +NO$_2$ reaction, N$_2$O$_5$ uptake in NBL,
and N$_2$O$_5$ uptake in RL/N$_2$O$_5$ uptake from RL mixed process) at the GIG and Heshan
sites. The error bars represent the standard deviations of the mean production rate.

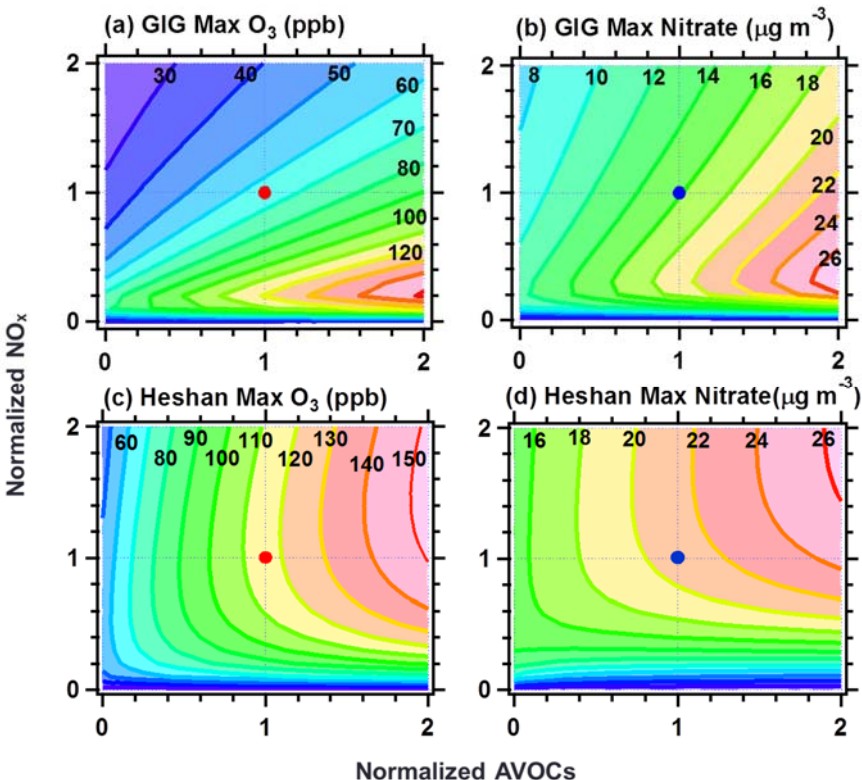

**Figure 10**. The simulated isopleths of ozone and nitrate with normalized $NO_x$ and AVOCs concentration at the (a, b) GIG site and (c, d) Heshan site, each isopleth represents the maximum ozone and nitrate in the simulation, and the red and blue circles represent the base cases.


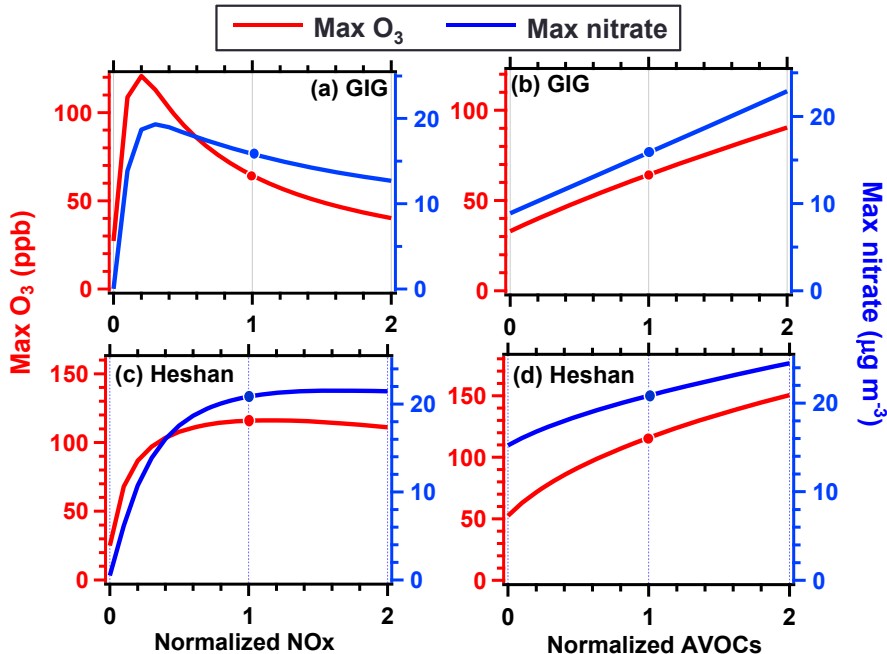


**Figure 11**. Simulated maximum ozone and nitrate concentration with normalized
NOx and AVOCs at the (a, b) GIG site and (c, d) Heshan site, cutting through the
simulated isopleth in Figure 8 with normalized AVOCs and NOx ratio at 1, respectively.
The red and blue circles represent the base cases.

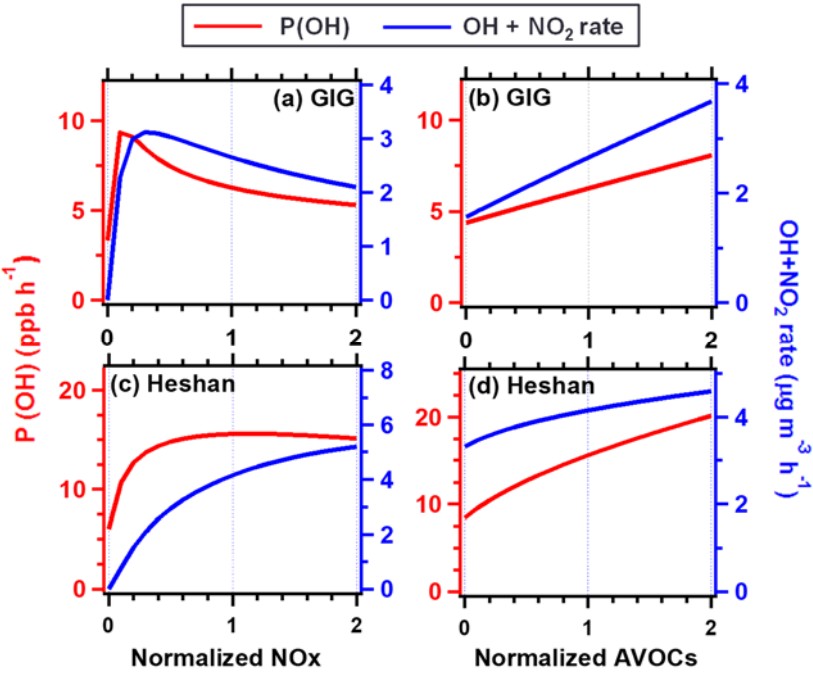


**Figure 12.** Simulated average production rates of OH (P (OH)) and the reaction rate of
OH and NO2 with the normalized changes of NOx and AVOCs emissions at the (a, b)
GIG site and (c, d) Heshan site.

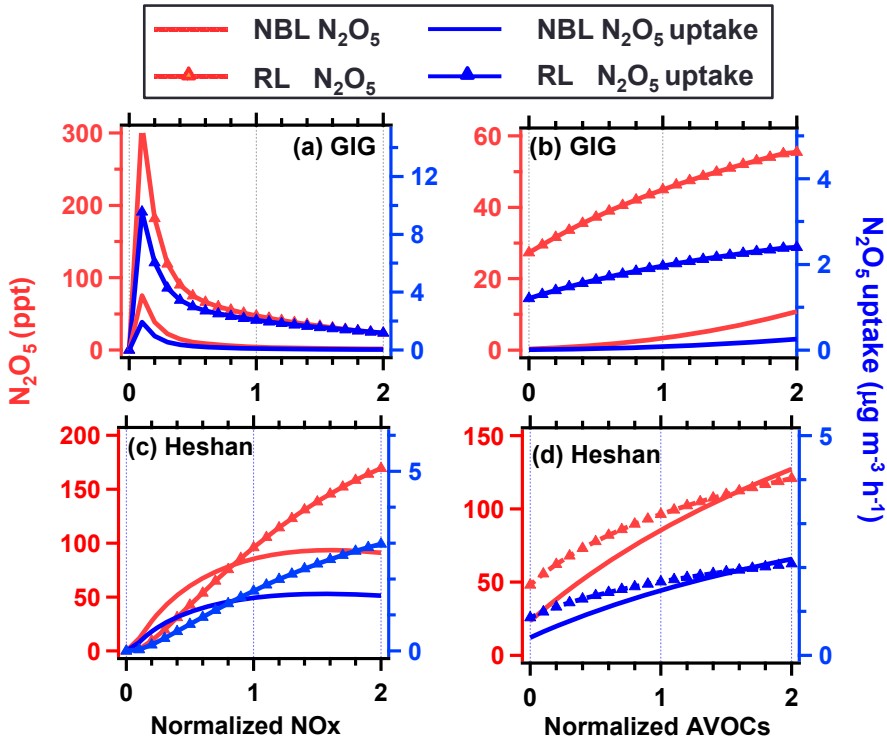


**Figure 13.** Simulated average concentration of $N_2O_5$ and nitrate production rate from
$N_2O_5$ uptake with the normalized changes of $NO_x$ and AVOCs emissions at the (a, b)
GIG site and (c, d) Heshan site in the NBL and RL.