# Peer review of "The formation and mitigation of nitrate pollution"

_Atmospheric Chemistry and Physics, 2021_

## Referee Comment (RC1)

Review "The formation and mitigation of nitrate pollution: Comparison between urban and suburban environments"

**General Remarks:**
The authors investigated the formation processes of nitrate in both urban and suburban areas using local chemical and meteorological measurements in southern China and a chemical box model. They found that reducing nitrate is essential for reducing the occurrence of aerosol pollution since the higher ratios of nitrate/sulfate occurred during the polluted periods. They further explored the relevant key factors in nitrate chemistry and concluded that it is necessary to integrate emission controls on $NO_x$ and VOCs to have a comprehensive mitigation of nitrate pollution over different environments. This is an interesting and valuable study. I recommend publishing it in ACP after the authors make the following major modifications.

**Major comments:**
It may be worth adding one more simulation at the GIG station with a similar methodology adopted at the HeShen, i.e., allowing ground measured chemical fields to evolve freely. There are two advantages to this approach: 1). Such a simulation provides a clean comparison between GIG and HeShen for their chemical evolution in NBL and RL, and such a comparison is key for this study. 2). The evaluation of this new simulation against their conducted GIG simulation (i.e., using the observed tower-level data) targets the end nitrate products directly, not only at a couple of important tracers as shown in Fig S6.

I am concerned with the roles of $SO_4^{2-}$ and $NH_3$ in influencing $NO_3^-$ chemistry. For example, even though the molar ratios of measured $[NH_4^+]$ to the sum of $2 \times [SO_4^{2-}] + [NO_3^-]$ are close to 1.0 at the two sites, that does not mean the regions have sufficient $NH_3$ to further neutralize nitrate as those experiments shown in Figs 8-9. One potential test is to check whether doubling $NH_3$ emissions yields doubling predicted $NO_3^-$.

It would be good to explain the diffusive time scale used. Is the lifetime of 24 h applied to every species or only for the secondary species? How sensitive is it to the simulation results? Is there any evidence or reference for the chosen lifetime?

I also feel it is hard to follow the description of the measurement at the three sites and of the experiment set up. The authors might consider summarizing the relevant important information in tables. Clarification is also needed of several definitions used in the text, such as residual layer, transition regime, update rate. See more details in specific comments.

**Specific comments:**
1. Page 3 lines 64-65: The increase of NH3 is also a reason.
2. Page 3 line 77: The chemistry processes described here are for inorganic nitrate only. Thus, the term "Particulate nitrate" should be "Particulate inorganic nitrate" to be more accurate.
3. Page 3 lines 84-85: Please check Table 1 and the corresponding discussion for the gamma values used in the nine global chemistry models in Bian et al., (2017).
4. Page 4 line 93: Please define "the residual layer" here.

5. Page 4 line 111: How do you account for the role of SO4 in adjusting this thermodynamic equilibrium reaction?

6. Pages 5-7 lines 144-192: A table that summarizes important information for the three measurement sizes would be helpful.

7. Page 5 line 156: When was the tower-based measurement conducted?

8. Page 6 lines 177-182: Why use different instruments? Have you calibrated the two instruments at the same time and location?

9. Page 6 lines 180-182: Same question.

10. Page 6 line 189: "campaign" should be "campaigns"? Otherwise please indicate which campaign.

11. Page 7 line 206-209: What if the simulation of the residual layer at the GIG site also freely evolved from sunset time using the ground observation data?

12. Page 8 lines 227-228: How do you get these y.

13. Page 8 lines 233-236: How large could the uncertainty in the simulated particulate nitrate be with this approach?

14. Page 8 line 239: Is it the overall aerosol liquid water or the liquid water for different aerosol compositions? Could the authors elaborate a bit more on the f(RH) calculation?

15. Page 9 line 266: What do the authors mean by "N2O5 constrained"? Do you refer to inclusion of the heterogeneous uptake of N2O5? Or does it mean to assimilate observed N2O5 in the calculation?

16. Page 9 lines 267-268: Please describe more for the base case setup. Could the authors give a table to summarize the designed simulations? The authors pointed out, in lines 265-266, that the base case is the one without N2O5 constrained. Thus, I am confused with the sentence here: "The base case simulation was comparable to the results without N2O5 constrained".

17. Page 9 lines 275-280: The authors discussed the various potential uncertainties in simulation, but how about the uncertainty of the diffusion adopted in the study?

18. Page 11 lines 315-317: I do not understand this sentence.

19: Page 11 lines 322-325: These lines seem to describe the performance of GIG and should be moved ahead before the discussion of HeShan, i.e., before the sentence in line 319 starting with "At the Heshan site, …".

20. Page 12 line 348: I do not think "comparable" is the right word. The simulated N2O5 in HeShan is significantly higher than that of measurement.

21. Page 12 lines 356-358: Here, the authors pointed out that it is not necessary to use observed N2O5 constraining for the nitrate simulation. However, the authors also pointed out that the nighttime uptake of N2O5 is important for the nitrate simulation. How do you reconcile these points?

22. Page 13 lines 388-389: Is there any data to support this conclusion?

23. Page 13 line 411-412: What is the N2O5 uptake rate? Is it the reaction rate defined in Eq. 1? If yes, please keep the same name throughout the paper. If not, please define it and describe the method of its calculation.

24. Page 14 line 417: Is it the column or surface to the "total nitrate production"?

25. Page 14 line 444: Please define "the transition regime" here.

26. Page 16 line 491-494: What was the "the titration effect of NO on NO3 radical and ozone at the GIG site" and why did it not occur at the HeShan site?

27: Page 17 line 523: It might be more appropriate to use the word "difference" instead of "opposite" here.

28: Page 18 line 545: What are these limitations and how large could their possible impact be on the study?
29. S Page 4 line 78: Was there any precipitation occurred during the studied period?

**Technique corrections:**
1. Page 4 line 109: Please define $\varphi$ in the paragraph right after this equation.
2. Page 7 line 221: Please define Sa in the paragraph right after this equation, similar to w1 and r.
3. Page 8 line 222: Please define $J_{ClNO2}$.
4. Page 8 line 223: Please give the unit to w1.
5. Page 8 line 224: Please delete the definition of $\varphi$ here.
6. Page 8 line 236: Please move the definition of Sa to line 224.
7. Page 8 line 249: Please give the unit to Ra.
8. Page 8 line 250: Please give the unit to w.
9. Page 17 line 514: Please define "PRD".
10. Page 14 Figure S3c: Please change label "Hehan $N_2O_5$" to "HeShan $N_2O_5$".

Bian, H., Chin, M., Hauglustaine, D. A., Schulz, M., Myhre, G., Bauer, S. E., Lund, M. T., Karydis, V. A., Kucsera, T. L., Pan, X., Pozzer, A., Skeie, R. B., Steenrod, S. D., Sudo, K., Tsigaridis, K., Tsimpidi, A. P., and Tsyro, S. G.: Investigation of global nitrate from the AeroCom Phase III experiment, Atmos. Chem. Phys., 17, 12911-12940, https://doi.org/10.5194/acp-17-12911-2017, 2017.

---

## Referee Comment (RC3)

The authors describe a recent field campaign at an urban, suburban, and tower measurement site near Guangzhou, China. They use these observations to construct a box model for the production of nitrate aerosol, and demonstrate that the urban area is in a VOC-limited regime, while the suburban site is at a transition point. The tower measurements yield critical information about the contribution of different production mechanisms in the nocturnal boundary and residual layers.

Overall, this is a very good paper that provides new constraints on an important pollution issue, and I recommend publication. I have only a few minor comments.

General comments: Would the authors include more details about what (if any) biogenic VOCs are included in the model.

Line 82 – Previous work has emphasized the importance of particle pH in nitrate aerosol formation, so this should be discussed at some point. See Guo, H., Otjes, R., Schlag, P., Kiendler-Scharr, A., Nenes, A., and Weber, R. J.: Effectiveness of ammonia reduction on control of fine particle nitrate, Atmos. Chem. Phys., 18, 12241-12256, 2018 as an example.

Line 134 – What is meant by "different environments"? The authors should be a little more clear about what makes this paper different than other recent papers discussing NOx and VOC sensitivity in urban areas in China.

Line 155 – Change "upward" to "upwind"

Line 157 – It's not clear here whether the tower measurements were taken during the same timeframe as the GIG ground site.

Line 164 – Were the aethelometer and particle size distributions taken at the GIG site? If so, change line 157 to read "The chemical components of PM1, trace gases, NMHC, and particle BC content and size were measured…."

Line 196 – A reference detailing the MCM should be cited here.

Line 229 – State what the observed parameters were.

Line 373 – Where does the estimate of the nocturnal boundary layer and residual layer fractions as 0.4 / 0.6 come from? Is this an empirical observation during the study or an estimate based on theory?

Figure 4: I would suggest putting the modeled diurnal observations on the observation to make the comparison more clear.

---

## Author Comment (AC1)

**Reply to the comments of Anonymous Referee #1**

**General Remarks:**

The authors describe a recent field campaign at an urban, suburban, and tower measurement site near Guangzhou, China. They use these observations to construct a box model for the production of nitrate aerosol, and demonstrate that the urban area is in a VOC-limited regime, while the suburban site is at a transition point. The tower measurements yield critical information about the contribution of different production mechanisms in the nocturnal boundary and residual layers.

Overall, this is a very good paper that provides new constraints on an important pollution issue, and I recommend publication. I have only a few minor comments.

Reply: We thank the reviewer for the comments. These comments are valuable and very helpful for improving this paper. We reviewed these comments carefully and made corresponding revisions according to the reviewer's comments. Our replies to the comments are itemized below in blue color.

**General comments:**

1. Would the authors include more details about what (if any) biogenic VOCs are included in the model.

Reply: We thank the reviewer for the comment. Isoprene was included as other researchers done in the box models (Tan et al., 2018). To clarify this issue, we have added the corresponding descriptions in **line 294~297** in the revised manuscript as follows.

**"Isoprene was included in the simulation as biogenic VOC (BVOC). Reducing BVOCs such as isoprene is impractical, so it is not scaled with AVOCs concentrations in the sensitivity simulations on control of precursors."**

2. Line 82 – Previous work has emphasized the importance of particle pH in nitrate aerosol formation, so this should be discussed at some point.

See Guo, H., Otjes, R., Schlag, P., Kiendler-Scharr, A., Nenes, A., and Weber, R. J.:

Effectiveness of ammonia reduction on control of fine particle nitrate, Atmos. Chem. Phys., 18, 12241-12256, 2018 as an example.

Reply: We agree with your comment that pH plays an important role in the nitrate formation by affecting the thermal equilibrium and gas-particle partitioning. We have added the sentences in **line 86~88** in the revised manuscript and cited this paper as you have suggested.

**"The pH value within a certain range plays an important role in the gas-particle partitioning of nitrate, which significantly impacts the nitrate formation (Guo et al., 2018;Lawal et al., 2018;Nenes et al., 2020)"**

3. Line 134 – What is meant by "different environments"? The authors should be a little more clear about what makes this paper different than other recent papers discussing NOx and VOC sensitivity in urban areas in China.

Reply: The "different environments" means different emission ratios of NOx and VOCs in ambient atmosphere, such as urban and suburban sites. The nitrate formation impacted by the NOx-VOCs-O$_3$ chemistry was evaluated in this study, which combined ground- and tower-based measurements to simulate the nitrate formation aloft at urban and suburban sites. This issue has not been systematically evaluated in reported field studies. To address this issue clearly, we have modified the "different environments" to **"urban and suburban areas"** in **line 154** in the revised manuscript as follows.

**"In addition, few studies have comprehensively evaluated the relative influence of NOx and VOCs reductions on nitrate production in the urban and suburban areas."**

4. Line 155 – Change "upward" to "upwind"

Reply: We modified the "upward" to **"upwind" in line 175** in the revised manuscript.

5. Line 157 – It's not clear here whether the tower measurements were taken during the same timeframe as the GIG ground site.

Reply: The tower measurements were taken during the same period as the GIG

ground site, we have added the field measurement period in **line 176 ~ 179** in the revised manuscript.

**"The tower-based measurements were conducted simultaneously at the ground and 448 m on the Canton Tower from late September to mid-November in 2018 concurrent with the measurements at the GIG site, which are approximately 5.7 km apart each other."**

6. Line 164 – Were the aethelometer and particle size distributions taken at the GIG site? If so, change line 157 to read "The chemical components of PM1, trace gases, NMHC, and particle BC content and size were measured…."

Reply: Yes, the BC and particle size distribution were measured at the GIG site. We have revised this sentence in **line 180** in the revised manuscript as you have suggested.

**"The chemical components of $PM_1$, trace gases, and non-methane hydrocarbons (NMHC), and particle BC content and particle size distribution were both measured at the GIG and Heshan sites, whereas only trace gases (NOx and $O_3$) and meteorological parameters were measured at the Canton Tower site".**

7. Line 196 – A reference detailing the MCM should be cited here.

Reply: We thank the reviewer for the suggestion. Some references which described the MCM in detail were cited in the revised manuscript as follows.

Bloss, C., Wagner, V., Jenkin, M. E., Volkamer, R., Bloss, W. J., Lee, J. D., Heard, D. E., Wirtz, K., Martin-Reviejo, M., Rea, G., Wenger, J. C., and Pilling, M. J.: Development of a detailed chemical mechanism (MCMv3.1) for the atmospheric oxidation of aromatic hydrocarbons, Atmos. Chem. Phys., 5, 641-664, 10.5194/acp-5-641-2005, 2005.

Jenkin, M. E., Saunders, S. M., Wagner, V., and Pilling, M. J.: Protocol for the development of the Master Chemical Mechanism, MCM v3 (Part B): tropospheric degradation of aromatic volatile organic compounds, Atmos. Chem. Phys., 3, 181-193, 10.5194/acp-3-181-2003, 2003.

Saunders, S. M., Jenkin, M. E., Derwent, R. G., and Pilling, M. J.: Protocol for the

development of the Master Chemical Mechanism, MCM v3 (Part A): tropospheric degradation of non-aromatic volatile organic compounds, Atmos. Chem. Phys., 3, 161-180, 10.5194/acp-3-161-2003, 2003.

The revisions have been made in **line 218~222** are as follows:

**"The F0AM box model uses a subset of the Master Chemical Mechanism (MCM) v3.3.1 (Saunders et al., 2003;Jenkin et al., 2003;Bloss et al., 2005), which explicitly describe chemical reactions of VOCs, ROx radicals (including OH, HO$_2$ and RO$_2$), ozone and nitrate, and was widely used in laboratory and theoretical researches (Edwards et al., 2017;Anderson et al., 2017;D'Ambro et al., 2017;Womack et al., 2019). "**

8. Line 229 – State what the observed parameters were.

Reply: The observed mean data of γ at the Heshan site, combined with flow-tube system, was $0.020 \pm 0.019$. We have revised the sentence in **line 251~ 253** in the revised manuscript as follows.

**"The average values of γ were 0.018 ± 0.01 and 0.019 ± 0.01 at the GIG and Heshan sites, respectively, which were comparable with the observed mean data of γ (0.020 ± 0.019) at the Heshan site in 2017 (Yu et al., 2020)."**

9. Line 373 – Where does the estimate of the nocturnal boundary layer and residual layer fractions as 0.4 / 0.6 come from? Is this an empirical observation during the study or an estimate based on theory?

Reply: The PBL height data were derived from the NOAA Air Resource Laboratory website (https://ready.arl.noaa.gov/READYamet.php). The average diurnal boundary layer height was 400 m and 1000 m in the nighttime and daytime during the study period, respectively, which are shown in **Fig. S1** in the revised manuscript as follows. Thus, the heights of the nocturnal boundary layer and residual layer were set as 400 m and 600 m, and the nocturnal boundary layer and residual layer fraction was estimated as 0.4 and 0.6, respectively.

[Figure]

**Figure S1. Diurnal variations of mean Planetary Boundary Layer (PBL) heights at (a) GIG site and (b) Heshan site, which were obtained from the NOAA Air Resource Laboratory website (https://ready.arl.noaa.gov/READYamet.php); (c) Schematic of PBL evolution and chemistry in the box model.**

10. Figure 4: I would suggest putting the modeled diurnal observations on the observation to make the comparison more clear.

Reply: We thank the reviewer for the suggestion. We have added the modeled and observed diurnal nitrate concentrations in **Fig.S10** in the revised manuscript, and described the comparison in **line 414 ~ 419 of Page 14** in the revised manuscript as follows.

**"The diurnal simulated nitrate was comparable with the observation at the GIG site, especially when considering the vertical transport from the residual layer in the morning. Unlike the GIG site, the diurnal simulated nitrate performed higher in the daytime, and little bit lower in the late nighttime, compared with the observation. It may be related to the lack of quantitative transport in the box model."**

[Figure]

**Figure S10.** **Comparison of daily-averaged box model simulated and observed nitrate at the GIG and Heshan site**

**References**

[revised manuscript text omitted]

---

## Author Comment (AC2)

**Reply to the comments of Anonymous Referee #2**

Referee comment on "The formation and mitigation of nitrate pollution: Comparison between urban an suburban environments" by Suxia Yang et al., Atmos. Chem.Phys.Discuss., https://doi.org/10.5194/acp-2021-730-RC2, 2021

**General comments**

Yang *et al.* analyze data from several sites in the Pearl River Delta to assess mechanisms for the production of nitrate aerosol, an increasingly important component of PM2.5 pollution in China. The analysis shows that the contribution of photochemical and dark mechanisms varies by site and depends on both the chemistry and the dynamics of the planetary boundary layer. It further shows that NOx reductions are unlikely to improve nitrate pollution despite being the major precursor due to the dependence of NOx oxidation rates on NOx itself. Reductions in VOCs, by contrast, are effective at all sites in both NOx and O₃ reductions. The paper is well written, easy to follow and well organized. It is of substantial interest to the readership of ACP. I recommend publication following attention to the specific comments below.

Reply: We would like to thank the reviewer for the insightful comments, which help us tremendously in improving the quality of our work. Please find the responses to individual comments below.

**Specific comments**

1. Line 53: Nitrate reductions can be site specific, but the same is true for ozone and for the same reasons as detailed later in the manuscript. Can identify this effect here.

Reply: We thank the reviewer for the suggestion. The ozone production and synergistic control of nitrate and ozone pollution are also site-specific. "The results highlight that the relative importance of nitrate formation pathways can be site-specific, and the quantitative understanding of various pathways of nitrate formation will provide insights for developing nitrate mitigation strategies." has been revised in **line 53 ~ 55** in the revised manuscript as follow.

**"The results highlight that the relative importance of nitrate and ozone formation can be site-specific, and the quantitative understanding of various pathways of**

**nitrate formation will provide insights for developing nitrate and ozone mitigation strategies."**

2. Line 71-74: Nitrate photolysis to produce HONO remains uncertain. References that also place limits on this process should be included.

Romer, P.S., *Constraints on Aerosol Nitrate Photolysis as a Potential Source of HONO and NOx.* Environmental Science & Technology, 2018. **52**(23): p. 13738-13746.

Reply: We thank the reviewer for the valuable suggestion. Indeed, the photolysis of particulate nitrate to produce HONO had some limitations and uncertainties, as the enhancement of particulate nitrate photolysis may not be so fast. Thus, we rephrased the sentence in **line 71 ~76** in the revised manuscript as follows.

**"In addition, the photolysis of particulate nitrate can increase the production of sulfate and nitrous acid (HONO), implying the importance of nitrate in the synergetic enhancement of the atmospheric oxidizing capability in haze events (Gen et al., 2019;Zhang et al., 2020;Ye et al., 2016;Ye et al., 2017), although the photolysis of particulate nitrate to produce HONO still remains highly uncertain (Romer et al., 2018)."**

3. Lines 81-84: Aerosol pH is also an important process that should be identified and referenced for $HNO_3$ partitioning. See for example:

Guo, H., *Effectiveness of ammonia reduction on control of fine particle nitrate.* Atmos. Chem. Phys., 2018. **18**(16): p. 12241-12256.

Lawal, A.S., *Linked Response of Aerosol Acidity and Ammonia to SO2 and NOx Emissions Reductions in the United States.* Environmental Science & Technology, 2018. **52**(17): p.9861-9873.

Nenes, A., *Aerosol pH and liquid water content determine when particulate matter is sensitive to ammonia and nitrate availability.* Atmos. Chem. Phys., 2020. **20**(5): p. 3249-3258.

Franchin, A., *Airborne and ground-based observations of ammonium-nitrate-dominated aerosols in a shallow boundary layer during intense winter pollution*

*episodes in northern Utah.* Atmos. Chem. Phys., 2018. **18**(23): p. 17259-17276.

Reply: We agree with your comment that pH plays an important role in the nitrate formation by affecting the thermal equilibrium and gas-particle partitioning. We have reorganized the sentence in **line 83 ~ 88** in the revised manuscript as follows.

**"The partitioning process of HNO$_3$ between the gas and particle phase is regulated by ambient temperature (T), relative humidity (RH) (Mozurkewich, 1993), aerosol pH and the abundance of NH$_3$ (R2) (Xue et al., 2014;Yun et al., 2018;Franchin et al., 2018). The pH value within a certain range plays an important role in the gas-particle partitioning of nitrate, which significantly impacts the nitrate formation (Guo et al., 2018;Lawal et al., 2018;Nenes et al., 2020)."**

4. Line 87: Can also reference McDuffie 2018b for the variation of ClNO$_2$ yields.

McDuffie, E.E., *ClNO2 Yields From Aircraft Measurements During the 2015 WINTER Campaign and Critical Evaluation of the Current Parameterization.* Journal of Geophysical Research: Atmospheres, 2018. **123**(22): p. 12,994-13,015.

Reply: We thank the reviewer for the suggestion. We have added this reference in **line 110** in the revised manuscript.

5. Line 95: Also suggest earlier references from California, e.g.

Brown, S.G., *Wintertime Vertical Variations in Particulate Matter (PM) and Precursor Concentrations in the San Joaquin Valley during the California Regional Coarse PM/Fine PM Air Quality Study.* Journal of the Air & Waste Management Association, 2006. **56**(9): p.1267-1277.

Chow, J.C., *PM2.5 chemical composition and spatiotemporal variability during the California Regional PM10/PM2.5 Air Quality Study (CRPAQS).* Journal of Geophysical Research: Atmospheres, 2006. **111**(D10): p. n/a-n/a.

Reply: We thank the reviewer for the valuable suggestion. We have added these references in **line 121** in the revised manuscript.

5. Line 190-191: There is a reference to integrity and temporal coverage of the measurements as a limitation on the data, without much explanation. More detail on which instruments were functioning at which times could be given in the introduction to this section or the SI.

Reply: Considering the different measurement time between the instruments, we choose the overlapping time as the study period. To clarify this issue, we have added the temporal coverage of different instruments at GIG site, Ganton Tower site and Heshan site in **Table S1~ Table S3** in the revised manuscript as follows.

**Table S1. Measured chemical species and the analytical methods, time resolution, limit of detection, the accuracy of the instruments used for different measured species, and sampling period at the GIG site.**

| chemical species | methods | time resolution | limit of detection | accuracy | sampling period |
|---|---|---|---|---|---|
| NMHC | GC-FID-MS | 1 h | $10 \sim 84$ ppt | $0.65\% \sim 9.14\%$ | 2018.09.14~2018.11.19 |
| Formaldehyde | PTR-TOF-MS | 1 min | 20 ppt | 11.80% | 2018.09.12~2018.11.19 |
| Acetaldehyde | PTR-TOF-MS | 1 min | 33 ppt | 12.50% | 2018.09.12~2018.11.19 |
| MVK+MACR | PTR-TOF-MS | 1min | 8 ppt | 5.8% | 2018.09.12~2018.11.19 |
| $HNO_3$ | TOF-CIMS | 1 min | < 10 ppt | $\pm 20\%$ | 2018.10.07~2018.11.19 |
| $N_2O_5$ | TOF-CIMS | 1 min | < 10 ppt | $\pm 25\%$ | 2018.10.07~2018.11.19 |
| $ClNO_2$ | TOF-CIMS | 1 min | < 10 ppt | $\pm 25\%$ | 2018.10.07~2018.11.19 |
| $NH_3$ | CRDS | 1 min | 1.0 ppb | $\pm 35\%$ | 2018.09.30~2018.10.29 |
| HONO | LOPAP | 1 min | 6.0 ppt | $\pm 20\%$ | 2018.09.28~2018.11.19 |
| $O_3$ | UV absorption | 1 min | 0.5 ppb | $\pm 10\%$ | 2018.09.11~2018.11.20 |
| $NO/NO_2/NO_x$ | Chemiluminescence | 1 min | 0.4 ppb | $\pm 10\%$ | 2018.09.11~2018.11.20 |
| CO | Infrared absorption | 1 min | 0.04 ppm | $\pm 10\%$ | 2018.09.11~2018.11.20 |
| $NO_3^-$, $SO_4^{2-}$, $NH_4^+$ | TOF-AMS | 300 s | $0.005 \sim 0.024$ $\mu g\ m^{-3}$ | $\pm 20\%$ | 2018.09.29~2018.11.20 |
| Sa | APS (500 nm to 20 μm), SMPS (10 to 650 nm) | 300s | — | $\pm 10\%$ | 2018.09.29~2018.11.20 |
| Photolysis frequencies | Spectrometer | 10 s | — | $\pm 10\%$ | 2018.09.18~2018.11.19 |

**Table S2. Measured chemical species and the analytical methods, time resolution, limit of detection, the accuracy of the instruments used for different measured species, and sampling period at the Canton Tower site.**

| Location | chemical species | methods | time resolution | limit of detection | accuracy | sampling period |
|---|---|---|---|---|---|---|
| Ground site and 488 m site | $O_3$ | UV absorption | 1 min | 0.5 ppb | ± 10% | 2018.09.20~2018.11.20 |
| | $NO/NO_2/NO_x$ | Chemiluminescence | 1 min | 0.4 ppb | ± 10% | 2018.09.20~2018.11.20 |
| | CO | Infrared absorption | 1 min | 0.04 ppm | ± 10% | 2018.09.20~2018.11.20 |

**Table S3. Measured chemical species and the analytical methods, time resolution, limit of detection, the accuracy of the instruments used for different measured species, and sampling period at the Heshan site.**

| chemical species | methods | time resolution | limit of detection | accuracy | sampling period |
|---|---|---|---|---|---|
| NMHC | GC-FID-MS | 1 h | $0.01 \sim 0.41$ ppb | _ | 2019.09.25~2019.11.16 |
| Formaldehyde | PTR-TOF-MS | 1 min | 29 ppt | 15.6% | 2019.10.16~2019.11.16 |
| Acetaldehyde | PTR-TOF-MS | 1 min | 18 ppt | 4.2% | 2019.10.16~2019.11.16 |
| MVK+MACR | PTR-TOF-MS | 1min | 7.3 ppt | 5% | 2019.10.16~2019.11.16 |
| $HNO_3$ | TOF-CIMS | 1 min | < 10 ppt | ± 20% | 2019.10.01~2019.11.16 |
| $N_2O_5$ | TOF-CIMS | 1 min | < 10 ppt | ± 25% | 2019.10.01~2019.11.16 |
| $ClNO_2$ | TOF-CIMS | 1 min | < 10 ppt | ± 25% | 2019.10.01~2019.11.16 |
| $NH_3$ | GAC | 30 mins | 0.08 ppb | _ | 2019.09.25~2019.11.16 |
| HONO | GAC | 30 mins | 0.1 ppb | _ | 2019.09.25~2019.11.16 |
| $O_3$ | UV absorption | 1 min | 0.5 ppb | ± 10% | 2019.09.25~2019.11.16 |
| $NO/NO_2/NOx$ | Chemiluminescence | 1 min | 0.4 ppb | ± 10% | 2019.09.25~2019.11.16 |
| CO | Infrared absorption | 1 min | 0.04 ppm | ± 10% | 2019.09.25~2019.11.16 |
| $NO_3^-$, $SO_4^{2-}$, $NH_4^+$ | TOF-AMS | 300 s | $0.005\sim0.024$ $\mu g\ m^{-3}$ | ± 20% | 2019.10.02~2019.11.16 |
| Sa | APS (500 nm to 20 μm), SMPS (10 to 650 nm) | 300s | — | ± 10% | 2019.10.02~2019.11.16 |
| Photolysis frequencies | Spectrometer | 10 s | — | ± 10% | 2019.09.28~2019.11.16 |

6. Line 216: How was the dilution rate determined? This is an important parameter that is normally fit to achieve agreement with observations in box modeling approaches. The 24 hour inverse rate constant appears to be rather an arbitrary guess.

Reply: We thank the reviewer for the constructive comments. The box model

cannot replicate the effects of meteorology, and the dilution is not accurately quantitated. A "physical loss" lifetime of 6 h ~ 48 h is used to prevent long-lived species to build-up (Wolfe et al., 2016). The empirical lifetime of 24 h was often used in box model, or determined combining with the performance of OVOCs and target species (Lu et al., 2017;Decker et al., 2019;Zhao et al., 2020;Novak and Bertram, 2020;Souri et al., 2020;Liu et al., 2021). A lifetime of 8 h was used in the study of nitrate formation at the Heshan site in 2017 by Yun et al. (2018). We did the sensitivity tests combined with the unconstrained OVOCs species (MVK+MACR), $O_3$, $HNO_3$ and nitrate at the GIG and Heshan site, as shown in **Fig.S2 and Fig.S3** in the revised manuscript as follows. A dilution rate of 24 $h^{-1}$ for all species was determined at the GIG site, due to the reasonable consistency between the simulation and observation for the chosen species. Comparing with the diurnal average observation, the average variations were -19% for MVK + MACR, -10% for $O_3$, 25% for $HNO_3$, and 12% for nitrate with the dilution constant of 8 $h^{-1}$ at the Heshan site. The simulated diurnal data had relative minor deviation with the observation by dilution constant of 8 $h^{-1}$. Thus, the lifetime of 24 h and 8 h were used at the GIG and Heshan site, respectively. We have clarified these issues in the revised manuscript and SI as follows.

[Figure]

**Figure.S2 Sensitivity tests with different dilution constant (kdilution) at the GIG site by box model.**

[Figure]

**Figure.S3 Sensitivity tests with different dilution constant (kdilution) at the Heshan site by box model.**

Line 241~244 in the Method of revised manuscript:

"To prevent the build-up of long-lived species to unreasonable levels, an additional physical dilution process was applied in the model (Lu et al., 2017;Decker et al., 2019;Novak and Bertram, 2020;Liu et al., 2021;Yun et al., 2018). To achieve agreement with the observation, a life time of 24 h and 8 h were used at the GIG and Heshan site, respectively. The sensitivity tests were shown in Fig.S2 and Fig.S3."

Line 79~89 in the SI:

"The physical loss that parameterized as a first order dilution process was the same as the daytime simulation in both the NBL and RL. We did the sensitivity tests combined with the unconstrained OVOCs species (MVK+MACR), $O_3$, $HNO_3$ and nitrate at the GIG and Heshan site, as shown in Fig.S2 and Fig.S3. A dilution rate of 24 $h^{-1}$ for all species was determined at the GIG site, due to the good consistency between the simulation and observation for the chosen species. Comparing with the diurnal average observation, the simulated diurnal data had relative minor deviation with the observation by dilution constant of 8 $h^{-1}$. The

**average variations were -19% for MVK + MACR, -10% for O$_3$, 25% for HNO$_3$, and 12% for nitrate with the dilution constant of 8 h$^{-1}$ at the Heshan site. Thus, the dilution constant of 24 h$^{-1}$ and 8 h$^{-1}$ were used at the GIG and Heshan site, respectively."**

7. Line 264-265: Explain why this approach is meaningless.

Reply: The N$_2$O$_5$ production was affected by the O$_3$ and NO$_2$, which would change with the VOCs and NOx emission. If we constrain N$_2$O$_5$ as the observation, the concentration of N$_2$O$_5$ would keep constant as NOx or VOCs concentrations change in different simulation scenarios, which could not provide the feedback of nocturnal nitrate formation due to the precursors change. In addition, the modeled results without N$_2$O$_5$ constrained were comparable to the observations. Thus, we did not constrain N$_2$O$_5$ in the simulation. We have rephrased it in the revised manuscript in **line 298 ~ 300** in the revised manuscript as follows.

**"Since the N$_2$O$_5$ is affected by the chemistry between ozone and VOCs, constraining N$_2$O$_5$ concentrations with the change in NOx ratio arbitrarily during the isopleth simulations is improper."**

8. Lines 271-273: Large N$_2$O$_5$ mixing ratios were present elsewhere in the time series in S3 but do not appear to be associated with poor representation of nitrate in S2. Is this explanation consistent with the data?

Reply: Indeed, large N$_2$O$_5$ mixing ratios were present elsewhere at the GIG site, such as the early nighttime in October 22 and late nighttime in October 26 in **Fig.S6** as follows in the revised manuscript (original Fig.S3); and the simulated nitrate with N$_2$O$_5$ constrained were also higher than the base case at that time in **Fig.S5** as follows in the revised manuscript (original Fig.S2). Besides, the observed nitrate increased with the high concentration of N$_2$O$_5$. It is consistent with the results. The simulated nitrate with N$_2$O$_5$ constrained on October 9 and October 10 were significantly higher, so we specially pointed out the simulation on these days. The GIG site is located near traffic avenues, impacted by traffic emissions during the nighttime. The abnormally high observed concentrations of N$_2$O$_5$ lasted for short periods (10-30 minutes) at the

nighttime, which may be caused by the transported air masses from upwind regions or vertical transport without well-mixed with fresh urban NO emissions. The high $N_2O_5$ in short period may be related to the short lifetime of $N_2O_5$ (Brown et al., 2006b) , which will be investigated in the future.

[Figure]

**Figure S5.** Comparison of the box model simulated and observed nitrate concentrations at (a) GIG site and (b) Heshan site. The orange lines represent simulated results of the base case (S0) without $N_2O_5$ constrained, and green lines represent the simulated results with $N_2O_5$ constrained (S1).

[Figure]

**Figure S6**. Comparison of the box model simulated and observed N₂O₅ and ClNO₂ concentrations at (a, b) GIG site and (c, d) Heshan site.

9. Line 289-294: Suggest comparing this result to that of the Franchin paper above, which shows the same effect but more dramatically for aircraft data in northern Utah, USA.

Reply: As you suggested, we have compared the results with those from Franchin's paper, and have added the corresponding sentences in **Page 11 line 329 ~ 330** in the revised manuscript.

**"The significant increasing ratio of nitrate fraction from clean condition to polluted condition (~ 43%) was also revealed in the airborne observations in Utah Valley, US (Franchin et al., 2018)."**

10. Line 309-311: Meaning of this sentence is not clear. Is the morning increase in nitrate being attributed to photochemical NO₂ oxidation in the residual layer, or does

the word "might" here indicate uncertainty? If the former, the later discussion of nighttime accumulation of nitrate would appear to conflict with this statement.

Reply: We thank the reviewer for the comment. We inferred the increase of nitrate was induced by the aloft transportation, so the word "might" indicate uncertainty. At the GIG site, with the rapid increase of nitrate during morning time, the precursors of $HNO_3$ and $NH_3$ did not increase sharply. $HNO_3$ mainly formed by the photochemical reaction of $OH + NO_2$. So, this process might be caused by the vertical transport from aloft. In order to clarify this issue, we have rephrased the sentence (original line 309 ~ 311) "The concentration of $NO_2$ exhibited a decreasing trend during the nitrate growth period. As gaseous $HNO_3$ is mainly produced by the reaction of OH and $NO_2$, the accumulation of nitrate after sunrise might largely be attributable to the downward transport from the residual layer to the ground." in **Page 11 line 344 ~ 348** in the revised manuscript as follows.

**"At the GIG site, nitrate rapidly increased from 4:00 to 9:00, but the concentrations of $NH_3$ and $HNO_3$ increased slowly, which suggests the minor contribution of direct production of $HNO_3$ from the reaction of OH and NO2. The increase of nitrate during this period might be associated with the downward transport from the residual layer to the ground."**

11. Line 316: A sustained level of nearly 2 ppbv of NO in excess $O_3$ at 488 m implies very rapid mixing with surface NO emissions. Is this likely to be the case, and if so, would it be consistent with an analysis of an isolated residual layer? More likely might be that the NO instrument zero is not well characterized, and that NO was in fact zero at this altitude. If so, the reaction of $NO_3 + NO$ would present no limit for nighttime chemistry at 488 m.

Reply: We thank the reviewer for the valuable comments. The observed NO concentration was between 1 ~ 3 ppb at the 488 m site on the Canton tower when $O_3$ concentration was almost 50 ppb during the nighttime, as shown in the following Fig.R1 (a). However, the NO concentration was nearly zero in the nighttime when $O_3$ was high at both GIG and Heshan site (Fig.R1 (b ~ c)). We compared the nighttime NO concentration against $O_3$ at the GIG site, Heshan site and 488 m site of Canton Tower.

When the $O_3$ concentration was between 50 ppb to 60 ppb during the nighttime, the average concentration of NO was 0.8 ppb, 0.05 ppb and 2.1 ppb at the GIG site, Heshan site and 488 m site of Canton Tower, respectively. In addition, the modeled concentration of NO was also shown in Fig.R2, which was simulated by freely evolved NO at the residual layer. The average simulated concentration of NO was 0.02 ppb when the $O_3$ was between 50 ppb to 60 ppb.

As pointed by the reviewer, we agree that the NO measurements at the 488 m on the Caton tower may be somewhat biased. Actually, it is a common issue for NO measurements at continuously monitoring stations in China, as the result of infrequently instrumental background measurements. Thus, NO concentrations should be close to zero at the 488 m site of Canton Tower, rather than 1-3 ppb based on the measurement.

It should be noted that the potential NO measurement errors do not affect the modelling results in this study. In the modelling of the residual layer for the urban region, only $NO_2$ and $O_3$ were constrained using measurements at the 488 m site of Canton Tower and NO was freely evolved (see details in Text S1 in SI).

[Figure]

Figure R1 Time series of observed O₃ (left) and NO (right) concentration at the (a) 488 m site of Canton Tower, (b) GIG site and (c) Heshan site. The shadows at the 488 m site of Canton Tower represent the stage that the concentration of O₃ was high, while NO still kept a higher concentration.

[Figure]

Figure R2 scatter plot between nighttime NO and O₃ at GIG site (blue blocks), Heshan site (orange blocks), at 488 m site of Canton Tower (light blue and green blocks). The green blocks represent the modeling concentration of NO at 488 m site of Canton Tower

We have updated the description about NO concentration at the 488 m site of Canton Tower **in line 351 ~ 354** in the revised manuscript as follows.

**"However, the average concentration of O₃ at 488 m of Canton Tower site was 2.4 times higher than that at the GIG site during nighttime, and the lower nocturnal concentrations of NO (nearly zero) at the 488 m site would enhance the production of NO₃ and N₂O₅ (Wang et al., 2018b;McDuffie et al., 2019)."**

12. Line 336: The instrument descriptions indicate that NH₃ was measured. Was there excess gas phase NH₃ as implied by the ion balance in Figure S5?

Reply: We thank the reviewer for the insightful comment. We also calculated the excess NH₃ by molar concentration of Total ammonia ([NH₃] +[NH₄⁺]) - Total nitrate([HNO₃] + [NO₃⁻]) -2*sulfate(2*[SO₄²⁻]) (Seinfeld &Pandis et al., 2008), as shown in the following Fig. R3. The average excess ammonia at the GIG and Heshan

site were 0.18 $\mu$mol m$^{-3}$ and 0.05 $\mu$mol m$^{-3}$, respectively, indicating excess gas NH$_3$ at the two sites.

[Figure]

Figure R3. The excess ammonia calculated by molar concentration of Total ammonia ([NH$_3$] +[NH$_4^+$] - Total nitrate([HNO$_3$] + [NO$_3^-$]) -2*sulfate(2*[SO$_4^{2-}$]) at the GIG and Heshan site.

13. Line 351-353: The effect of periodic large N$_2$O$_5$ and ClNO$_2$ is more likely due to vertical than horizontal transport – so these concentrations may be associated with the overlying residual layer.

Reply: We thank the reviewer for the comment. Indeed, the periodic large N$_2$O$_5$ and ClNO$_2$ may be associated with vertical transport from residual layer. In general, when the residual layer decoupled from the nocturnal layer, the vertical mixing is weak since the residual layer is one neutral stable layer. Due to lacking of radar or related vertical observation, we did not emphasize the vertical transport effect of periodic large N$_2$O$_5$ and ClNO$_2$. We considered your suggestion and modified the sentence in **Page 14 line 424 ~ 427** in the revised manuscript.

**"The abnormally high observed concentrations of N$_2$O$_5$ and ClNO$_2$ that lasted for short periods (10-30 minutes) at the GIG site may be caused by**

**transported air masses from upwind regions or vertical transport without well-mixed with fresh urban NO emissions."**

14. Line 430: The model of residual and boundary layer mechanisms for nitrate production is certainly more complete than most similar analyses. However, horizontal transport in the residual layer, especially as part of nocturnal jets, has been invoked in some analyses of winter nitrate production in the California central valley (see Brown and Chow references above). Some comment in this section about the differences in horizontal transport would be useful, even if it is not possible to quantitatively analyze this effect for the data in this study. The assumption here is that the residual layer and the nocturnal boundary layer originate at the same location, which is not necessarily the case. As noted later in the paper, this is one of the limitations of box modeling.

Reply: We thank the reviewer for the constructive comments. The horizontal transport occurred in the residual layer, which would facilitate the formation and transportation of nitrate ammonium in regional scale. As you have mentioned in the comment that it is not possible to conduct horizontal simulation via box model, thus, we have added the related implications and suggestions on the possibility of horizontal transport to the nitrate production in the residual layer. The revisions are shown in **Page 16 line 509 ~ 513** in the revised manuscript as follows.

**"The horizontal transport in the residual layer from nocturnal jets may also contribute to the different nitrate production at urban and suburban sites, which has been discussed in the research of Chow et al. (2006) and Brown et al. (2006a). Due to the limitation of box model, this issue could be studied by the chemistry transport model in further research."**

15. Line 445-446: The NOx sensitivity at Heshan looks neutral or near peak – that is $O_3$ and nitrate would stay approximately constant for an initial NOx reduction. Also, could define what is meant by "initial' here – just an infinitesimal increment, or a fixed number such as 5 or 10%.

Reply: We thank the reviewer for the comment. It is true that the maximum ozone and nitrate concentration show little decrease for an initial NOx reduction. The maximum ozone decreased from 116 ppb to 115.3 ppb, and the maximum nitrate

concentration decreased from 20.8 μg m$^{-3}$ to 20.5 μg m$^{-3}$ with 10% reduction of NOx. With the increased reduction ratio of NOx, the decrease rate of ozone and nitrate enhanced. Thus, the nitrate and ozone at the Heshan site were in the transition regime. A less than 70% reduction of NOx emission would increase nitrate and ozone concentrations at the GIG site, an "initial" reduction meant the reduction ratio before the turning point of O$_3$ or nitrate appeared. To clarify this issue, we have rephrased the sentence **in Page 17 line 527 ~ 530** in the revised manuscript.

**"As shown in Fig. 11, the reduction of NOx emissions from 0-70% would increase nitrate and ozone concentrations at the GIG site, but decrease those concentrations at the Heshan site. The decrease in VOCs concentrations would decrease nitrate and ozone concentrations at both sites."**

Supplement

16. Lines 135-138: The sensitivity to the ClNO$_2$ yield is explored, but not the N$_2$O$_5$ uptake coefficient. Can the authors comment on the sensitivity to this parameter? Importantly, there may be almost no sensitivity here if the system is limited by the reaction of NO$_2$ + O$_3$. If so, the N$_2$O$_5$ uptake coefficient would need to be reduced substantially before the heterogeneous reaction becomes important or rate limiting. Canthe authors comment on these aspects of the model sensitivity?

Reply: We thank the reviewer for the suggestion. Indeed, parameterized N$_2$O$_5$ uptake parameter ($\gamma$) is also an important parameter for nocturnal chemistry, and changes widely between laboratory and field studies. We used the median parameterized $\gamma$ and $\varphi$ as the base input, and changed the two parameters respectively to perform the sensitivity tests. The results showed that the nitrate sensitivity did not change with different values of $\gamma$ and $\varphi$. It is consistent with the research by Womack et al. (2019) in the Salt Lake of US. When $\gamma$ is in a certain range, the nocturnal nitrate production is limited by the formation of NO$_3$ and N$_2$O$_5$ (Chen et al., 2020;McDuffie et al., 2019). We have revised these sentences in **line 157 ~ 163** in the revised SI manuscript as shown below, and updated the **Fig.S14** (original Fig.S9) in the SI manuscript.

**"Here we chose the median value of $\gamma$ (0.018) and $\varphi$ (0.18) as the base input**

parameters; thus, different values of γ and φ were selected to perform sensitivity simulation (Fig. S14). Compared with the base case, the sensitivity of nitrate did not change with different values of γ and φ although the peak values of nitrate showed difference with the changing of γ and φ".

[Figure]

**Figure S14. Sensitivity tests of the production yield of ClNO₂ (φ value) and the uptake parameter of N₂O₅ (γ value) on maximum nitrate concentrations as a function of the normalized NOx and AVOCs relative to the base concentration at the GIG site.**

**Technical corrections**

1. Line 43: replace "are" with "is an"

Reply: We have replaced the "are" with **"is an" in line 43** in the revised manuscript.

2. Line 62: hygroscopic properties

Reply: We have changed the word "hygroscopic property" to **"hygroscopic properties" in line 63** in the revised manuscript.

3. Line 231: particle rather than particles

Reply: We have revised the word "particles" to **"particle" in line 255** in the revised manuscript.

4. Line 365-367: Check sentence grammar

Reply: We have rephrased the sentence (original line 365 ~ 367) "Since the nitrate produced in the residual layer is only gradually mixed to the surface as the boundary layer develops during the following morning, while the nitrate contributed to the boundary layer column concentration always included the $N_2O_5$ uptake in the residual layer during the whole nighttime (Wang et al., 2018a;Womack et al., 2019)." **in line 439 ~ 443** in the revised manuscript.

**"The nitrate produced in the residual layer is only gradually mixed to the surface as the boundary layer develops during the following morning, while the nitrate contributed to the boundary layer column concentration always included the $N_2O_5$ uptake in the residual layer during the whole nighttime"**

5. Line 399-400: Check meaning – what is "nitrate of nitrate"

Reply: We have changed the original sentence (original line 399 ~ 400) "The relative magnitudes of the contributions to the daily-averaged surface nitrate of nitrate differ somewhat from the contributions to the entire boundary layer." in **line 477** in the revised manuscript as follows.

[revised manuscript text omitted]

---

## Author Comment (AC3)

**Reply to the comments of Anonymous Referee #3**

[Atmospheric Chemistry and Physics, MS ID: acp-2021-730]

Title: The formation and mitigation of nitrate pollution: Comparison between urban and suburban environments

**General Remarks:**

The authors investigated the formation processes of nitrate in both urban and suburban areas using local chemical and meteorological measurements in southern China and a chemical box model. They found that reducing nitrate is essential for reducing the occurrence of aerosol pollution since the higher ratios of nitrate/sulfate occurred during the polluted periods. They further explored the relevant key factors in nitrate chemistry and concluded that it is necessary to integrate emission controls on NOx and VOCs to have a comprehensive mitigation of nitrate pollution over different environments. This is an interesting and valuable study. I recommend publishing it in ACP after the authors make the following major modifications.

Reply: We thank the reviewer for the comments. These comments are valuable and very helpful for revising and improving our paper, as well as the important guiding significance to our study. We reviewed these comments carefully and made necessary revisions according to the reviewer's comments. Our Replies to the comments are itemized below in blue color.

**Major comments:**

1. It may be worth adding one more simulation at the GIG station with a similar methodology adopted at the HeShen, i.e., allowing ground measured chemical fields to evolve freely. There are two advantages to this approach: 1). Such a simulation provides a clean comparison between GIG and HeShan for their chemical evolution in NBL and RL, and such a comparison is key for this study. 2). The evaluation of this new simulation against their conducted GIG simulation (i.e., using the observed tower-level

data) targets the end nitrate products directly, not only at a couple of important tracers as shown in Fig S6.

Reply: We thank the reviewer's suggestion. Actually, the simulation suggested by the reviewer which treated the GIG site with a similar methodology at the Heshan site, allowing the ground measured chemical fields to evolve freely, has been already included in the original manuscript (original Fig.S6). These results were presented in **Fig.S11** in the revised manuscript as follows. The simulated results of trace gases ($NO_2$, $NO_x$, $O_3$ and $O_x$) agreed well with the observed data at the 488m Canton Tower site. Based on the reliability of the test simulation, we modeled the residual layer chemistry at the Heshan site.

1) The simulated comparison about nitrate production rate from $N_2O_5$ uptake between GIG and Heshan with the same methodology is shown in the following Fig. R4 (a, b), indicating the chemical evolution in the nocturnal layer and residual layer. The nitrate production rate from $N_2O_5$ uptake in the residual layer was significantly higher than that in the nocturnal layer at the GIG site, suggesting active nocturnal chemistry in the residual layer. It was consistent with the analysis in the manuscript which suggested the importance of the nitrate production from $N_2O_5$ uptake in the residual layer at the GIG site. At the Heshan site, the nitrate production rate from $N_2O_5$ uptake in the residual layer was comparable with that in the nocturnal layer, with no significant difference between the two layers.

[Figure]

Figure.R4 Time series of simulated nitrate production rate from $N_2O_5$ uptake in the nocturnal layer and residual layer at the GIG and Heshan site with the same methodology, which adopted ground observation data as the initial input in the residual layer.

2)We have added the modeled nitrate and nitrate production rate from $N_2O_5$ uptake with the observation from the GIG and 488m Canton Tower site as initial input in the residual layer in **Fig.S11 (e) ~ (f)** of the revised manuscript as follows. The results agreed well with each other. We have mentioned this point in **line 75 ~ 78** in the revised **Text S1** in SI as follows.

**"In addition, the simulated nitrate and nitrate production rate from $N_2O_5$ uptake with the observation at the GIG site and 488 m Canton Tower as initial input were also compared in Fig.S11 (e ~ f), showing good agreement. Thus, we adopted this simulation method to perform the simulations at the Heshan site."**

[Figure]

**Figure S11.** Time series of the simulated trace gases ((a) $NO_2$, (b) $NO_x$, (c) $O_3$ and (d) $O_x$) in the RL, when the observations at 17:00 at GIG were setting as the initial inputs of the RL simulation and all chemical species were freely evolved in the box model. (e) the simulated nitrate and (f) nitrate production rate from $N_2O_5$ uptake with the observation data at the GIG (black line) and 488m Canton Tower (blue line) as the initial inputs in the RL. The observations at GIG and 488m site of Canton Tower are also shown for comparison. The error bars represent the standard deviation of the observations.

2. I am concerned with the roles of $SO_4^{2-}$ and $NH_3$ in influencing $NO_3^-$ chemistry. For example, even though the molar ratios of measured $[NH_4^+]$ to the sum of $2\times[SO_4^{2-}]+[NO_3^-]$ are close to 1.0. at the two sites, that does not mean the regions have sufficient $NH_3$ to further neutralize nitrate as those experiments shown in Figs 8-9. One potential test is to check whether doubling $NH_3$ emissions yields doubling predicted

$NO_3^-$.

Reply: We thank the reviewer for the insightful comment. Generally, $H_2SO_4$ is firstly neutralized by $NH_3$ to form sulfate due to its lower saturated vapor pressure. Then, the rest of $NH_3$ will react with $HNO_3$ to produce ammonium nitrate. The influence of sulfate and $NH_3$ on the nitrate production both lies on the available of $NH_3$ to neutralize nitrate. Thus, the test mentioned by you are needed to clarify the $NH_3$ effect on the nitrate formation.

Here, we use the thermodynamic ISORROPIA II model to evaluate the effect of $NH_3$ and sulfate in the nitrate formation. The same evaluation has been reported in the studies of Guo et al. (2018) and Nenes et al. (2020), which both suggest the particle fraction of nitrate in the sum of $HNO_3$ and nitrate ($\varepsilon(NO_3^-)$) was affected by the pH values. In this work the ISORROPIA II model is based on the available input of total gas and particulate matter ($HNO_3$ + nitrate, $NH_3$ + ammonium, sulfate, and chloride), T and RH, and is run in the "forward" and "metastable" mode. The ISORROPIA II modeled results of nitrate, ammonium, $HNO_3$, and $NH_3$ at the GIG and Heshan site were displayed in **Fig.S8 ~ Fig.S9** in the revised manuscript as follows. The modeled components from ISORROPIA II showed good correlations with the observed concentrations at both sites.

The results of experiment by increasing total ammonium (NHx, ammonium + $NH_3$) by 50% and 100%, keeping other parameters constant, are listed in Table R1. When doubling the total ammonium, the predicted nitrate increased 25% compared with the nitrate in the base case, suggesting doubling $NH_3$ would benefit the production of nitrate but not in a linear increase. This reveals both $NH_3$ and $HNO_3$ are important precursors for nitrate. It did not affect the experiments shown in original Figs 8-9. As $NH_3$ plays an important role in the nitrate partitioning, we further evaluated the effect of $NH_3$ and sulfate reduction by thermodynamic ISORROPIA II model in this study.

Table R1 ISORROPIA II predicted average nitrate and relative changes

| ISORROPIA model case | predicted average nitrate($\mu$ g m$^{-3}$) | Relative changes |
| --- | --- | --- |
| Base case | 9.1 | – |
| + 50% NHx | 10.7 | + 18% |
| + 100% NHx | 11.4 | + 25% |

[Figure]

**Figure S8. Scatter plot of observations vs ISORROPIA II modeled results of nitrate, ammonium, HNO₃ and NH₃ at the GIG site during the study period.**

[Figure]

**Figure S9. Scatter plot of observations vs ISORROPIAII modeled results of nitrate, ammonium, HNO₃ and NH₃ at the Heshan site during the study period.**

The effect of ammonium on nitrate partitioning is related with the pH value. Thus, the aerosol pH at the GIG and Heshan site is also calculated. The aerosol pH, based on

the aerosol acidity and water content, is calculated by the following equation:

$$pH = -log_{10} \frac{1000\, H_{air}^{+}}{ALWC}$$

where $H_{air}^{+}$ (µg m$^{-3}$) is the hydronium concentration of the equilibrium particle and ALWC (µg m$^{-3}$) is the aerosol water content from ISORROPIA II modeled results.

In the ISORROPIA base model, the fraction of nitrate ($\varepsilon(NO_3^-)$) in the sum of HNO$_3$ + nitrate against pH at the GIG and Heshan site are depicted in **Fig.6** in the revised manuscript as follows. The pH data are colored by relative humidity and fit to an "s-curve", as shown in Guo et al. (2018). The clustering of $\varepsilon(NO_3^-)$ data, mainly located between the pH values of 1~3, was sensitive to the changes in pH, and therefore may be sensitive to the changes of NH$_3$ and sulfate.

In the reduction cases, the input total ammonium (NHx, ammonium + NH$_3$) and sulfate were reduced from 10% to 90% relative to the ISORROPIA II base model, respectively, while keeping other parameters constant. The response of ISORROPIA II simulated nitrate concentration and aerosol pH to changes in NHx and SO$_4^{2-}$ are shown in **Fig.7** in the revised manuscript as follows. The nitrate concentration decreased with the reduction of NHx, and had little variation with the reduction of SO$_4^{2-}$ (Fig.7 (a ~ b)) at both sites. Along with the reduction of NHx, the pH values decreased significantly (Fig.7 (c ~ d)), which caused the further decrease of $\varepsilon(NO_3^-)$. The pH values showed a bit increase with the reduction of SO$_4^{2-}$, which may be caused by that there would be more available ammonium neutralized the hydronium. It is consistent with the study of Guo et al. (2018) and Nenes et al. (2020), which suggests the partitioning of nitrate was affected by the NH$_3$ in the pH values between 1~3. The partitioning of nitrate increased with the reduction of sulfate suggests the limited role of sulfate reduction on the mitigation of nitrate.

[Figure]

**Figure 6. The fraction of total nitrate that is partitioned to the particle phase ε(NO₃⁻) against aerosol pH. The pH data are colored by relative humidity and fit to an "s-curve" in black line, as shown in Guo et al. (2018).**

[Figure]

**Figure 7. ISORROPIA-predicted average nitrate (a, b) and pH (c, d) as a function of changes in NHx (ammonium + NH₃, orange line) and SO₄²⁻ (red line) at the GIG and Heshan site during the study period.**

The ISORROPIA II model setting was described in **Test S2** in the revised SI manuscript as follows. The influence of NH₃ reduction on the nitrate partitioning based on ISORROPIA II model results was added in **line 378 ~ 407** in the revised manuscript.

**"Text S2 Thermodynamic ISORROPIAII model description**

The presence of $HNO_3$ and $NH_3$ are conductive to form ammonium nitrate, which influenced by the aerosol pH and partitioning process of nitrate (Guo et al., 2018;Nenes et al., 2020). Thus, the thermodynamic ISORROPIA II model was used to evaluate the $NH_3$ and sulfate impacts on the gas-particle partitioning process of nitrate (Fountoukis and Nenes, 2007). The model is run in the "forward" and "metastable" mode, which is used to calculate the gas-particle equilibrium concentrations. The model is based on the available input of total gas and particulate measured matter ($HNO_3$ + nitrate, $NH_3$ + ammonium, sulfate, and chloride), T and RH. The low concentration of nonvolatile cations (such as sodium, calcium, potassium, magnesium) in the PRD region is assumed to have minor impact on the thermodynamic equilibrium in $PM_{1.0}$ (Franchin et al., 2018;Guo et al., 2018)."

We described the results of ISORROPIA II model in **Line 378 ~ 407** in the revised manuscript:

**"The ISORROPIA II model setting is described in Test S2 in detail. The ISORROPIA II modeled results of nitrate, ammonium, $HNO_3$, and $NH_3$ at the GIG and Heshan site were displayed in Fig.S8 ~ Fig.S9. The particle-phase nitrate and ammonium showed a bit overestimation at the GIG site, while the gas-phase $HNO_3$ and $NH_3$ showed overestimation at the Heshan site. Overall, the simulated components showed good correlations with the observed concentrations at both sites. We use the ISORROPIA II model results to evaluate the fraction of total nitrate that is partitioned to the aerosol phase $\varepsilon(NO_3^-)$ against aerosol pH. Aerosol pH, which depends on the aerosol acidity and water content, was calculated by the following equation:**

$$pH = -\log_{10} \frac{1000\ H_{air}^+}{ALWC} \qquad (Eq.5)$$

**where $H_{air}^+$ (µg m⁻³) is the hydronium concentration of the equilibrium particle and ALWC (µg m⁻³) is the aerosol water content from ISORROPIAII simulation.**

**The $\varepsilon(NO_3^-)$ against pH at the GIG and Heshan site are shown in Fig.6. The**

**pH data are colored by relative humidity and fit to an "s-curve" as in Guo et al. (2018). The clustering of pH data, mainly located between 1~ 3, and the ε(NO₃⁻) are sensitive to the change of pH. To further evaluate the sensitivity of NH₃ and sulfate on this effect, the input of total ammonium (NHx, ammonium + NH₃) and sulfate were reduced from 10% to 90% relative to the ISORROPIA II base model, respectively, while keeping other parameters constant. The response of ISORROPIA II simulated nitrate concentration and aerosol pH to changes in NHx and SO₄²⁻ are shown in Fig.7. The nitrate concentration decreased with the reduction of NHx, and had little variation with the reduction of SO₄²⁻ (Fig.7 (a ~ b)) at both sites. Along with the reduction of NHx, the pH values decreased significantly (Fig.7 (c ~ d)), which caused the further decrease of ε(NO₃⁻). The pH values showed a bit increase with the reduction of SO₄²⁻, which may be caused by that there would be more available ammonium neutralized the hydronium. It is consistent with the study of Guo et al. (2018) and Nenes et al. (2020), suggesting the partitioning of nitrate was also affected by the NH₃ in the pH values between 1~3. Thus, the control of NH₃ is effective for the reduction of nitrate by affecting the partitioning process of nitrate at both GIG and Heshan site in this study. The partitioning of nitrate increased with the reduction of sulfate suggests the limited role of sulfate reduction on the mitigation of nitrate."**

3. It would be good to explain the diffusive time scale used. Is the lifetime of 24 h applied to every species or only for the secondary species? How sensitive is it to the simulation results? Is there any evidence or reference for the chosen lifetime?

Reply: We thank the reviewer for the constructive comments. A "physical loss" lifetime of 6 h ~ 48 h is used to prevent long-lived species to build-up (Wolfe et al., 2016). The empirical lifetime of 24 h was often used in box model and determined through the combination with the performance of OVOCs and target species (Lu et al., 2017;Decker et al., 2019;Zhao et al., 2020;Novak and Bertram, 2020;Souri et al., 2020;Liu et al., 2021). A lifetime of 8 h was used in the study of nitrate formation at the Heshan site in 2017 by Yun et al. (2018). We did the sensitivity tests combined with the unconstrained OVOCs species (the sum of MVK + MACR), O₃, HNO₃ and nitrate at

the GIG and Heshan site, as shown in **Fig.S3 and Fig.S4** in the revised manuscript. A dilution rate of 24 h$^{-1}$ for all species was determined at the GIG site, due to the good consistency between the simulation and observation for the chosen species. Comparing with the diurnal average observation, the simulation diurnal data had relative minor deviation with the observation by dilution constant of 8 h$^{-1}$. The average variations were -19% for MVK + MACR, -10% for O$_3$, 25% for HNO$_3$, and 12% for nitrate with the dilution constant of 8 h$^{-1}$ at the Heshan site. Thus, the lifetime of 24 h and 8 h were used at the GIG and Heshan site, respectively. We have addressed this issue in the revised manuscript and SI as follows.

[Figure]

**Figure.S2 Sensitivity tests with different dilution constant (kdilution) at the GIG site by box model.**

[Figure]

**Figure.S3 Sensitivity tests with different dilution constant (kdilution) at the Heshan site by box model.**

Line 242~246 in the Method of revised manuscript:

"To prevent the build-up of long-lived species to unreasonable levels, an additional physical dilution process was applied in the model (Lu et al., 2017;Decker et al., 2019;Novak and Bertram, 2020;Liu et al., 2021). To achieve agreement with the observation, a life time of 24h and 8 h were used at the GIG and Heshan site, respectively. The sensitivity tests with different dilution constant at the GIG and Heshan site were shown in Fig.S2 and Fig.S3, respectively."

Line 79~89 in the revised SI:

"The physical loss that parameterized as a first order dilution process was the same as the daytime simulation in both the NBL and RL. We did the sensitivity tests combined with the unconstrained OVOCs species (MVK+MACR), $O_3$, $HNO_3$ and nitrate at the GIG and Heshan site, as shown in Fig.S2 and Fig.S3. A dilution rate of 24 $h^{-1}$ for all species was determined at the GIG site, due to the good consistency between the simulation and observation for the chosen species. Comparing with the diurnal average observation, the simulated diurnal data had relative minor deviation with the observation by dilution constant of 8 $h^{-1}$. The

**average variations were -19% for MVK + MACR, -10% for O$_3$, 25% for HNO$_3$, and 12% for nitrate with the dilution constant of 8 h$^{-1}$ at the Heshan site. Thus, the dilution constant of 24 h and 8 h was used at the GIG and Heshan site, respectively."**

4. I also feel it is hard to follow the description of the measurement at the three sites and of the experiment set up. The authors might consider summarizing the relevant important information in tables. Clarification is also needed of several definitions used in the text, such as residual layer, transition regime, update rate. See more details in specific comments.

Reply: We thank the reviewer for the useful suggestion. We have revised the measurement information at the three sites in **Table S1 ~ Table S3** separately in the revised manuscript, and summarized the important modeling information in **Table S5** in the revised manuscript as follows. Some definitions about residual layer, transition regime and uptake rate were revised in the manuscript, according to the reviewer's specific comments.

**Table S1. Measured chemical species and the analytical methods, time resolution, limit of detection, the accuracy of the instruments used for different measured species, and sampling period at the GIG site.**

| chemical species | methods | time resolution | limit of detection | accuracy | sampling period |
|---|---|---|---|---|---|
| NMHC | GC-FID-MS | 1 h | 10 ~ 84 ppt | 0.65% ~ 9.14% | 2018.09.14~2018.11.19 |
| Formaldehyde | PTR-TOF-MS | 1 min | 20 ppt | 11.80% | 2018.09.12~2018.11.19 |
| Acetaldehyde | PTR-TOF-MS | 1 min | 33 ppt | 12.50% | 2018.09.12~2018.11.19 |
| MVK+MACR | PTR-TOF-MS | 1min | 8 ppt | 5.8% | 2018.09.12~2018.11.19 |
| $HNO_3$ | TOF-CIMS | 1 min | < 10 ppt | $\pm$ 20% | 2018.10.07~2018.11.19 |
| $N_2O_5$ | TOF-CIMS | 1 min | < 10 ppt | $\pm$ 25% | 2018.10.07~2018.11.19 |
| $ClNO_2$ | TOF-CIMS | 1 min | < 10 ppt | $\pm$ 25% | 2018.10.07~2018.11.19 |
| $NH_3$ | CRDS | 1 min | 1.0 ppb | $\pm$ 35% | 2018.09.30~2018.10.29 |
| HONO | LOPAP | 1 min | 6.0 ppt | $\pm$ 20% | 2018.09.28~2018.11.19 |
| $O_3$ | UV absorption | 1 min | 0.5 ppb | $\pm$ 10% | 2018.09.11~2018.11.20 |
| $NO/NO_2/NO_x$ | Chemiluminescence | 1 min | 0.4 ppb | $\pm$ 10% | 2018.09.11~2018.11.20 |
| CO | Infrared absorption | 1 min | 0.04 ppm | $\pm$ 10% | 2018.09.11~2018.11.20 |
| $NO_3^-$, $SO_4^{2-}$, $NH_4^+$ | TOF-AMS | 300 s | 0.005~0.024 $\mu g\ m^{-3}$ | $\pm$ 20% | 2018.09.29~2018.11.20 |
| Sa | APS (500 nm to 20 $\mu$m), SMPS (10 to 650 nm) | 300s | — | $\pm$ 10% | 2018.09.29~2018.11.20 |
| Photolysis frequencies | Spectrometer | 10 s | — | $\pm$ 10% | 2018.09.18~2018.11.19 |

**Table S2. Measured chemical species and the analytical methods, time resolution, limit of detection, the accuracy of the instruments used for different measured species, and sampling period at the Canton Tower site.**

| Location | chemical species | methods | time resolution | limit of detection | accuracy | sampling period |
|---|---|---|---|---|---|---|
| Ground site and 488 m site | $O_3$ | UV absorption | 1 min | 0.5 ppb | ± 10% | 2018.09.20~2018.11.20 |
| | $NO/NO_2/NO_x$ | Chemiluminescence | 1 min | 0.4 ppb | ± 10% | 2018.09.20~2018.11.20 |
| | CO | Infrared absorption | 1 min | 0.04 ppm | ± 10% | 2018.09.20~2018.11.20 |

**Table S3. Measured chemical species and the analytical methods, time resolution, limit of detection, the accuracy of the instruments used for different measured species, and sampling period at the Heshan site.**

| chemical species | methods | time resolution | limit of detection | accuracy | sampling period |
|---|---|---|---|---|---|
| NMHC | GC-FID-MS | 1 h | $0.01 \sim 0.41$ ppb | _ | 2019.09.25~2019.11.16 |
| Formaldehyde | PTR-TOF-MS | 1 min | 29 ppt | 15.6% | 2019.10.16~2019.11.16 |
| Acetaldehyde | PTR-TOF-MS | 1 min | 18 ppt | 4.2% | 2019.10.16~2019.11.16 |
| MVK+MACR | PTR-TOF-MS | 1min | 7.3 ppt | 5% | 2019.10.16~2019.11.16 |
| $HNO_3$ | TOF-CIMS | 1 min | < 10 ppt | ± 20% | 2019.10.01~2019.11.16 |
| $N_2O_5$ | TOF-CIMS | 1 min | < 10 ppt | ± 25% | 2019.10.01~2019.11.16 |
| $ClNO_2$ | TOF-CIMS | 1 min | < 10 ppt | ± 25% | 2019.10.01~2019.11.16 |
| $NH_3$ | GAC | 30 mins | 0.08 ppb | _ | 2019.09.25~2019.11.16 |
| HONO | GAC | 30 mins | 0.1 ppb | _ | 2019.09.25~2019.11.16 |
| $O_3$ | UV absorption | 1 min | 0.5 ppb | ± 10% | 2019.09.25~2019.11.16 |
| NO/NO$_2$/NOx | Chemiluminescence | 1 min | 0.4 ppb | ± 10% | 2019.09.25~2019.11.16 |
| CO | Infrared absorption | 1 min | 0.04 ppm | ± 10% | 2019.09.25~2019.11.16 |
| $NO_3^-$, $SO_4^{2-}$, $NH_4^+$ | TOF-AMS | 300 s | $0.005{\sim}0.024$ µg m$^{-3}$ | ± 20% | 2019.10.02~2019.11.16 |
| Sa | APS (500 nm to 20 µm), SMPS (10 to 650 nm) | 300s | — | ± 10% | 2019.10.02~2019.11.16 |
| Photolysis frequencies | Spectrometer | 10 s | — | ± 10% | 2019.09.28~2019.11.16 |

**Table S5 Box model scenarios performed at the GIG and Heshan site**

| Site | Scenarios | description about simulation | Other information |
|---|---|---|---|
| GIG | base case (S0) | set lifetime as 24 h, without $N_2O_5$ constrained | NBL: with observation at the GIG site; RL: with Observation at the 488m site of Canton Tower |
|  | S1 | set lifetime as 24 h, with $N_2O_5$ constrained |  |
| Heshan | base case (S0) | set lifetime as 8 h, without $N_2O_5$ constrained | NBL: with observation at the Heshan site; RL: with observation at the Heshan site freely evolved |
|  | S1 | set lifetime as 8 h, with $N_2O_5$ constrained |  |

**Specific comments:**

1. Page 3 lines 64-65: The increase of $NH_3$ is also a reason.

Reply: We thank the reviewer for the useful comment. In the limited regulation of $NH_3$ emission, larger emission reduction of $SO_2$ than NOx was implemented in recent years, more free ammonia could react with gas $HNO_3$ (Guo et al., 2018;Liu et al., 2019;Zhai et al., 2021). Indeed, this is also one reason that the percentage of nitrate in $PM_{2.5}$ increased in recent years. We revised the original sentence (original line 64 ~ 65) "Due to the larger emission reduction of $SO_2$ than $NO_x$ was implemented since the clean air actions in China" in **Page 3 line 65** in the revised manuscript as follows.

**"Due to the larger emission reduction of $SO_2$ than NOx and little change of $NH_3$ since the implementation of the clean air actions in China (Guo et al., 2018;Liu et al., 2019;Zhai et al., 2021), a considerable increase in the nitrate fractions in aerosols has been observed in haze periods in the northern China Plain (Wen et al., 2018;Li et al., 2018;Lu et al., 2013;Fu et al., 2020), southern China (Pathak et al., 2009;Pathak et al., 2011) and eastern China (Griffith et al., 2015;Tao et al., 2018;Yun et al., 2018;Li et al., 2018), which indicates the growing significance of nitrate in the formation of haze events."**

2. Page 3 line 77: The chemistry processes described here are for inorganic nitrate only.

Thus, the term "Particulate nitrate" should be "Particulate inorganic nitrate" to be more accurate.

Reply: We have revised the term "Particulate nitrate" to "**Particulate inorganic nitrate**" in **Page 3 line 79** as follows.

**"Particulate inorganic nitrate is primarily produced through two processes: the photochemical reaction of hydroxyl radical (OH) and $NO_2$ during daytime (R1) and the heterogeneous uptake of $N_2O_5$ (R2–R5) during nighttime."**

3. Page 3 lines 84-85: Please check Table 1 and the corresponding discussion for the gamma values used in the nine global chemistry models in Bian et al., (2017).

Reply: We have checked the reference about **Bian et al., (2017),** and updated the references in **Page 4 line 109**.

4. Page 4 line 93: Please define "the residual layer" here.

Reply: We thank the reviewer for the comment. The nocturnal layer and residual layer are defined at the beginning of that paragraph in **Page 4 line 112 ~ 114** in the revised manuscript as follows.

**"With the radiative cooling in the afternoon, the mixed layer decoupled into a steady, near surface nocturnal boundary layer (NBL) and a residual layer (RL), which is a neutral layer and formed aloft during the turbulence attenuation process (Prabhakar et al., 2017)."**

5. Page 4 line 111: How do you account for the role of $SO_4^{2-}$ in adjusting this thermodynamic equilibrium reaction?

Reply: $H_2SO_4$ is preferred to be neutralized by ammonia to form sulfate, due to its lower saturated vapor pressure. The ammonia that does not react with sulfate will react with $HNO_3$ to form nitrate. We have evaluated the role of sulfate in the thermodynamic reaction by ISORROPIA II model in detail in the Major #2 comment.

6. Pages 5-7 lines 144-192: A table that summarizes important information for the three measurement sizes would be helpful.

Reply: We summarized the important information in **Table S1 ~ Table S3** in the revised manuscript as shown in Major # 4 comments.

7. Page 5 line 156: When was the tower-based measurement conducted?

Reply: The tower-based measurement was conducted from late September to mid-November in 2018 concurrently with the measurement at the GIG site, we have added this information in **Page 6 line 176~179** in the revised manuscript.

**"The tower-based measurements were conducted simultaneously at the ground and 448 m on the Canton Tower from late September to mid-November in 2018 concurrently with the measurements at the GIG site, which are approximately 5.7 km apart each other."**

8. Page 6 lines 177-182: Why use different instruments? Have you calibrated the two instruments at the same time and location?

Reply: We thank the reviewer for the comment. Because there was no available LOPAP instrument at the Heshan site, we measured HONO by GAC instrument at the Heshan site. The HONO concentrations measured by LOPAP and GAC have been compared in the southern and northern China by (Dong et al., 2012;Yang et al., 2014). The wet chemistry/ions chromatography method (as same as GAC method) had a better performance during the low HONO concentration (Xu et al., 2019;Xue et al., 2019;Cheng et al., 2013). The HONO concentration range was from 0.01 ppb ~ 2.2 ppb (mean value of 0.56 ppb) at the Heshan site. Thus, we used the HONO data measured by GAC instrument at the Heshan site.

9. Page 6 lines 180-182: Same question.

Reply: There was no CRDS instrument to measure $NH_3$ at the Heshan site during the campaign. The wet chemistry method (the same method as GAC instrument) and spectroscopic techniques (CRDS technique) agreed closely with each other in the comparing study of von Bobrutzki et al. (2010). Thus, we used the $NH_3$ data measured by GAC instrument at the Heshan site.

10. Page 6 line 189: "campaign" should be "campaigns"? Otherwise please indicate which campaign.

Reply: We have changed the "campaign" to **"campaigns"** in **Page 7 line 212** in the revised manuscript as follows.

**"The photolysis frequencies of $O_3$, $NO_2$, HCHO, and HONO (PFS-100,**

**Focused Photonics Inc., China) were also measured during the campaigns."**

11. Page 7 line 206-209: What if the simulation of the residual layer at the GIG site also freely evolved from sunset time using the ground observation data?

Reply: We thank the reviewer for the useful suggestion. The simulation suggested by the reviewer which treated the GIG site with a similar methodology at the Heshan site, allowing the ground measured chemical fields to evolve freely, has been already included in the original manuscript (original Fig.S6), as described in **Text S1** in the revised manuscript. The comparing results are shown in **Fig.S11** in the revised manuscript. The detailed reply was given in the Major #1 comment.

12. Page 8 lines 227-228: How do you get these y ($\gamma$).

Reply: The uptake parameter of $N_2O_5$ ($\gamma$) is calculated by the observation-based empirical parameterization method proposed in Yu et al. (2020) (Eq. R1), combined with chemical compositions of aerosol (nitrate and chloride), and the aerosol liquid water content (ALWC, [$H_2O$]). The nitrate and chloride were measured by the AMS instrument, and ALWC was calculated by the thermodynamic ISORROPIA model, as described in Franchin et al. (2018). The equations to determine $\gamma$ and $\varphi$ are:

$$\gamma_{N_2O_5} = \frac{4}{\omega_{N_2O_5}} * \frac{Va}{Sa} * K_H * 3.0e^4 * [H_2O] * (1 - \frac{1}{1+0.033*\frac{[H_2O]}{[NO_3^-]} + 3.4*\frac{[Cl^-]}{[NO_3^-]}}) \qquad \text{Eq. R1}$$

$$\varphi_{ClNO_2} = \frac{1}{1+\frac{0.033}{3.4}*\frac{[H_2O]}{[Cl^-]}} \qquad \text{Eq. R2}$$

Where $\omega_{N_2O_5}$ is the mean molecular speed of $N_2O_5$ (m s$^{-1}$), *Va/Sa* is the measured aerosol volume to surface aera ratio (m), $K_H$ is the Henry's law coefficient, [$H_2O$], [$NO_3^-$], and [$Cl^-$] are the aerosol water content, aerosol nitrate and chloride molarity (M), respectively.

13. Page 8 lines 233-236: How large could the uncertainty in the simulated particulate nitrate be with this approach?

Reply: As we shown in Eq. R1 and Eq. R2, the $N_2O_5$ uptake parameter($\gamma$) and $ClNO_2$ production yield($\varphi$)exhibited non-linear dependence on multiple factors, such as nitrate and organic matter concentration, aerosol liquid water content, chloride content and so on. Nitrate and organic matter can suppress the $N_2O_5$ uptake reaction in previous field studies, while higher RH and liquid water content likely promote $N_2O_5$

aqueous solvation and reaction (McDuffie et al., 2019). The $ClNO_2$ production yield is a function of the aerosol chloride to liquid water, with a negative dependence on aerosol liquid water and positive dependence on aerosol chloride (Bertram and Thornton, 2009;Yu et al., 2020). There was higher RH, and lower chloride at the 488 m site compared to the ground site of Canton Tower. The nitrate concentration was comparable at the 488 m site to the ground site in the study of Zhou et al. (2020). It is inferred that there are negative deviations for γ and positive deviations for φ in the residual layer, but the deviation may not be significant based on the results from McDuffie et al. (2018). The sensitivity tests about different γ and φ values have been included in **Fig.S14** in the revised manuscript as follows. When we reduced or increased the γ and φ by a factor of two, the relative deviations of modeled nitrate were less than 10%, compared to the base case at the GIG site. We revised the sentence in **Page 9 line 258 ~ 266** in the revised manuscript as follows.

**"The γ and φ exhibited complicated nonlinear dependence on aerosol composition, aerosol liquid water and RH (Bertram and Thornton, 2009;McDuffie et al., 2019;Yu et al., 2020), such that γ and φ has positive and negative dependence with RH, respectively. There was higher RH, and lower chloride at the 488 m site, compared to the ground site of Canton Tower. The nitrate concentration was comparable at the 488 m site to the ground site in the study of Zhou et al. (2020). Combined with the higher RH and lower $PM_{2.5}$ concentrations in the residual layer in this study (as shown in Fig.S4), we inferred the negative deviations for γ and positive deviations for φ in the residual layer."**

[Figure]

**Figure S14. Sensitivity tests of the production yield of ClNO₂ (φ value) and the uptake parameter of N₂O₅ (γ value) on maximum nitrate concentrations as a function of the normalized NOx and AVOCs relative to the base concentration at the GIG site.**

14. Page 8 line 239: Is it the overall aerosol liquid water or the liquid water for different aerosol compositions? Could the authors elaborate a bit more on the f(RH) calculation?

Reply: We thank the reviewer for the comment. The liquid water here means the inorganic-associated and organic-associated water in the aerosol. The inorganic-associated water was estimated from ISORROPIA II model, and the organic-associated water was estimated from the organic aerosol mass which was measured by AMS, and the organic hygroscopicity constant. To make it clear, we have revised the original sentence (original line 239) "f(RH) was calculated using the aerosol compositions measured by AMS and estimated liquid water by thermodynamic model of ISORROPIA, according to the study conducted by McDuffie et al. (2018)." in **Page 9 line 268 ~ 273** in the revised manuscript as follows.

**"The f(RH) was estimated from the aerosol composition measured by AMS and the aerosol liquid water content, which included the inorganic-associated and organic-associated water. The sum of inorganic-associated water estimated from**

**ISORROPIA thermodynamic model and organic-associated water estimated from the organic aerosol mass, was used to calculate the growth of wet matter contributions, as described in the study of McDuffie et al. (2018)."**

15. Page 9 line 266: What do the authors mean by "$N_2O_5$ constrained"? Do you refer to inclusion of the heterogeneous uptake of $N_2O_5$? Or does it mean to assimilate observed $N_2O_5$ in the calculation?

**Reply:** The "$N_2O_5$ constrained" means that the observed $N_2O_5$ was entered into the model in each modeled step. The $N_2O_5$ concentration at the beginning of each step is consistent with the observation.

16. Page 9 lines 267-268: Please describe more for the base case setup. Could the authors give a table to summarize the designed simulations? The authors pointed out, in lines 265-266, that the base case is the one without $N_2O_5$ constrained. Thus, I am confused with the sentence here: "The base case simulation was comparable to the results without $N_2O_5$ constrained".

**Reply:** We thank the reviewer for the suggestion. The base case (S0) is simulated without $N_2O_5$ constrained, and to test the results of base case, we design another simulation with $N_2O_5$ constrained (S1). The model scenarios were described in **Table S5** in the revised manuscript as follows. We have revised the legend in **Fig.S5** in the revised manuscript and the sentence in **Page 10 line 298 ~304** in the revised manuscript.

**Table S5 Box model scenarios performed at the GIG and Heshan site**

| Site | Scenarios | Description about simulation | Other information |
|------|-----------|------------------------------|-------------------|
| GIG | base case (S0) | set lifetime as 24 h, without $N_2O_5$ constrained | NBL: with observation at the GIG site; RL: with Observation at the 488m site of Canton Tower |
| | S1 | set lifetime as 24 h, with $N_2O_5$ constrained | |
| Heshan | base case (S0) | set lifetime as 8 h, without $N_2O_5$ constrained | NBL: with observation at the Heshan site; RL: with observation at the Heshan site evolved freely |
| | S1 | set lifetime as 8 h, with $N_2O_5$ constrained | |

**Page 10 Line 309 ~316** in the revised manuscript:

**"Since the $N_2O_5$ is affected by the chemistry between ozone and VOCs, constraining $N_2O_5$ concentrations with the change in NOx ratio arbitrarily during the isopleth simulations is improper. Thus, we set the simulation of base case (S0) without $N_2O_5$ constrained. To evaluate the results of the base case, we design another simulation with $N_2O_5$ constrained (S1) and compare the two simulated nitrate with the observation in Fig. S5. The model scenarios were described in Table S5 in detail. The base case simulation (S0) was comparable to the observation."**

17. Page 9 lines 275-280: The authors discussed the various potential uncertainties in simulation, but how about the uncertainty of the diffusion adopted in the study?

**Reply:** The concept of deposition rate meant the diffusion rate in the referenced studies (Lou et al., 2010;Lu et al., 2012). Considering the different diffusion rate adopted in this study, we evaluated the sensitivity of dilution rate in the **Fig.S2 and Fig.S3** in the revised manuscript, as described in the Major comment #3. In order to avoid inappropriate expression, we have rephrased the sentences in **Page 10 line 311 ~315** in the revised manuscript as follows.

**"Gaussian error propagation was used to evaluate the uncertainties about measurement parameters and reaction rates in the model, as described in Lu et al. (2012). The uncertainties of various measurement parameters (VOCs, trace gases, meteorological parameters, etc.) ranged from 0 to 20%, and uncertainties of reaction rates are in the order of ~20% (Lu et al., 2012)."**

18. Page 11 lines 315-317: I do not understand this sentence.

Reply: We emphasized that the difference of ozone and $NO_x$ concentration between the GIG site and the 488 m of Canton Tower site. The ozone concentration at the 488 m site was higher than the concentration at the GIG site, while NO concentration at the 488 m site was much lower than the concentration at the GIG site. The lower concentration of NO and higher concentration of ozone at the 488m site were favorable for the reaction of $NO_3$ and $N_2O_5$ in the residual layer. We have rephrased the

original line 315 ~ 317 "However, the average concentration of $O_3$ at 488 m was 2.4 times higher than that at the ground site during nighttime, and the low nocturnal concentrations of NO at the 488 m site would enhance the production of $NO_3$ and $N_2O_5$."in **Page 12 line 351 ~ 354** in the revised manuscript as follows.

**"However, the average concentration of $O_3$ at 488 m of Canton Tower site was 2.4 times higher than that at the GIG site during nighttime, and the lower nocturnal concentrations of NO (nearly zero) at the 488 m site would enhance the production of $NO_3$ and $N_2O_5$ (Wang et al., 2018;McDuffie et al., 2019)."**

19: Page 11 lines 322-325: These lines seem to describe the performance of GIG and should be moved ahead before the discussion of HeShan, i.e., before the sentence in line 319 starting with "At the Heshan site, …".

Reply: The original lines 322~325 describe the pollutants at the Heshan site, we have clarified it in **Page 12 line 358** in the revised manuscript.

**"Subsequently, there was a significant increase in nitrate from 7:00 to 9:00. The concentration of $NH_3$ showed variation pattern that was similar with that of nitrate and increased after 7:00, while the concentrations of $HNO_3$ and $NO_2$ showed a decreasing trend from 7:00 to 9:00 at the Heshan site."**

20. Page 12 line 348: I do not think "comparable" is the right word. The simulated N2O5 in HeShan is significantly higher than that of measurement.

Reply: We thank the reviewer for the suggestion. We agree with you that the simulated $N_2O_5$ and $ClNO_2$ at Heshan were higher than their observed concentrations, but the overall trend is consistent and comparable. We have modified the "comparable" as **"higher"** in **Page 14 line 425** in the revised manuscript as follows.

**"The temporal variations in simulated $N_2O_5$ and $ClNO_2$ concentrations were higher than the observations at the Heshan site as shown in Fig. S6 (c, d), the simulated results at the GIG site from October 9 to 10 were significantly lower than the observations (Fig.S6 (a, b))."**

21. Page 12 lines 356-358: Here, the authors pointed out that it is not necessary to use observed $N_2O_5$ constraining for the nitrate simulation. However, the authors also pointed out that the nighttime uptake of $N_2O_5$ is important for the nitrate simulation.

How do you reconcile these points?

Reply: We thank the reviewer for the comment. The simulated nitrate without $N_2O_5$ constrained (base case (S0)) was comparable with the observation, while the results with $N_2O_5$ constrained by S1 were significantly higher than observation on October 9 to 10 at the GIG site. The significantly higher nitrate by S1 may be caused by the upwind transported air masses without well-mixed with fresh urban NO emissions in short period (10 ~ 30 mins). Thus, the base case by S0 could represent the characterization of nitrate formation at the GIG site, suggesting the abnormal high concentration of $N_2O_5$ was negligible in short period. The well performance of modeled nitrate at the nocturnal and residual layer by S0 was based on the empirical calculation of the uptake coefficient of $N_2O_5$ and the production yields of $ClNO_2$, which also affected by the nocturnal chemistry in the model. In addition, the nighttime $N_2O_5$ uptake reaction was significant higher in the residual layer at the GIG site, which we emphasized the nighttime $N_2O_5$ uptake is important.

22. Page 13 lines 388-389: Is there any data to support this conclusion?

Reply: We thank the reviewer for the comment. The mean NO concentration in the early nighttime at the GIG site was 12.1 ppb. The lower concentration of NO (nearly zero) at the 488m site was favorable to produce $NO_3$ radical and $N_2O_5$. We have revised the sentence in **Page 15 line 466 ~ 468** in the revised manuscript.

**"This may be caused by the fresh NO surface emissions, which titrate the $NO_3$ radical and ozone in the nocturnal boundary layer, as the mean NO concentration in the nighttime at the GIG site was 12.1 ppb."**

23. Page 13 line 411-412: What is the $N_2O_5$ uptake rate? Is it the reaction rate defined in Eq. 1? If yes, please keep the same name throughout the paper. If not, please define it and describe the method of its calculation.

Reply: We thank the reviewer for the comment. The term "$N_2O_5$ uptake rate" is nitrate production rate from $N_2O_5$ uptake, which was defined in **Text S3** in the revised manuscript. The $k_6$ defined in Eq.1 is the reaction constant. To make it clear, we have changed the $N_2O_5$ uptake rate as **"nitrate production rate from $N_2O_5$ uptake"** in the revised manuscript.

24. Page 14 line 417: Is it the column or surface to the "total nitrate production"?

Reply: It means the contribution to the surface nitrate production; we have clarified it in **Page 16 line 494~497** in the revised manuscript.

**"The nitrate production rate from OH and NO$_2$ reaction (19.9 μg m$^{-3}$ day$^{-1}$) and nocturnal N$_2$O$_5$ uptake (15.6 μg m$^{-3}$ day$^{-1}$) were the major nitrate formation pathways, which contributed 56% and 44% to the surface total nitrate production, respectively."**

25. Page 14 line 444: Please define "the transition regime" here.

Reply: We have revised the sentence in **Page 17 line 527** in the manuscript.

**"The production of nitrate and ozone were in the VOCs-limited regime at the GIG site, and in the transition regime at the Heshan site, where nitrate and ozone are sensitive to both VOCs and NOx reduction."**

26. Page 16 line 491-494: What was the "the titration effect of NO on NO$_3$ radical and ozone at the GIG site" and why did it not occur at the HeShan site?

Reply: The NO consumes the NO$_3$ radical and ozone quickly during the higher NO concentration condition, which is called the titration effect. The ground surface average concentration of NO was 12.1 ppb and 0.5 ppb in the nighttime of GIG and Heshan site, respectively. As the lower NO concentration at the Heshan site, we did not emphasize the titration effect of NO. Another reason caused the N$_2$O$_5$ changes with NOx/VOCs emission was that ozone was in the NOx-saturation regime at the GIG site and in the transition regime at the Heshan site. It caused the initial ozone that participated in the nocturnal chemistry increase/decrease with the less than 70% NOx reduction at the GIG and Heshan site. We rephrased the sentences in **Page 18 line 576 ~ 580** in the revised manuscript.

**"During nighttime, the initial ozone concentration participated the nocturnal chemistry increased/decreased with the reduction of NOx at the GIG/Heshan site. In addition, the decrease in NOx will reduce the titration effect of NO on NO$_3$ radical and ozone at the GIG site, which enhances production of N$_2$O$_5$ and promotes nitrate production in both the nocturnal boundary layer and the residual layer (Fig.13)."**

27: Page 17 line 523: It might be more appropriate to use the word "difference" instead of "opposite" here.

Reply: We thank the reviewer for the suggestion. We have modified the word "opposite" to **"different"** in **Page 20 line 614~618** in the revised manuscript.

**"The spatial differences of nocturnal reactions and the different contributions from downward transport of the residual layer to surface nitrate at urban and suburban sites were attributed to different fresh emissions and concentration levels of NO$_x$ at the two sites during the night time, suggesting that nitrate production under different NO$_x$ conditions should be explored to better understand the its formation pathways."**

28: Page 18 line 545: What are these limitations and how large could their possible impact be on the study?

Reply: We thank the reviewer for the comment. The direct vertical transport and horizontal transport cannot be quantified in the 0-D box model, resulting in a lack of assessment about the transport contribution. Thus, three-dimensional models should be used to further investigate the synergistic control of ozone and particles on the regional scale.

We have added these limitations in **Page 20 line 635 ~ 638** in the revised manuscript.

**"As the result of limitation for the 0-D box model, vertical transport and horizontal transport cannot be considered explicitly in this study. Given the limitations of the box model, three-dimensional models should be used to further investigate the synergistic control of ozone and particles on the regional scale."**

29. S Page 4 line 78: Was there any precipitation occurred during the studied period?

Reply: There was mild rain on 10/15/2018~10/19/2018, and the precipitation intensity was 0.3 mm at the GIG site. At the Heshan site, there was no precipitation during the studied period. Thus, we did not consider the wet deposition in this study.

**Technique corrections:**

1. Page 4 line 109: Please define φ in the paragraph right after this equation.

Reply: We have defined φ in **Page 4 line 101** after the equation in the revised

manuscript.

2. Page 7 line 221: Please define Sa in the paragraph right after this equation, similar to w1 and r.

Reply: We have defined Sa, $\omega 1$ and$\gamma$in **Page 4 line 101 ~103** in the revised manuscript.

3. Page 8 line 222: Please define $JClNO_2$.

Reply: We have defined $JClNO_2$ in **Page 4 line 106** of the revised manuscript as follows.

**"Here the reaction rate k6 was denoted as the photolysis rate of $ClNO_2$ ($J_{ClNO_2}$)."**

4. Page 8 line 223: Please give the unit to w1.

Reply: The unit of $\omega 1$ is m s$^{-1}$, we have added the unit in **line 101** in the revised manuscript.

5. Page 8 line 224: Please delete the definition of $\varphi$ here.

Reply: We have removed the definition of $\varphi$ in the revised manuscript.

6. Page 8 line 236: Please move the definition of $S_a$ to line 224.

Reply: We have moved the definition of Sa to **line 102** in the revised manuscript.

7. Page 8 line 249: Please give the unit to Ra.

Reply: The unit of Ra is m, we have added the unit in the revised manuscript.

8. Page 8 line 250: Please give the unit to w.

Reply: The unit of $\omega$ is m s$^{-1}$, we have added the unit in the revised manuscript.

9. Page 17 line 514: Please define "PRD".

Reply: We have defined the Pearl River Delta as PRD in the Abstract.

10. Page 14 Figure S3c: Please change label "Hehan $N_2O_5$" to "HeShan $N_2O_5$".

Reply: We thank the reviewer for the reminder. We changed label "Hehan $N_2O_5$" to **"Heshan $N_2O_5$"**.

---

## Author Comment (AC4)

**Reply to the comments of Anonymous Referee #4**

This is a nice study to quantify the contributions of different formation mechanisms on nitrate at urban and suburban sites by using an observation-constrained box model. The authors found the important source of nitrate from the downwards transport of residual layer at the urban site, and a VOCs-limited chemical regime for nitrate formation, the nitrate formation was different at the suburban site. The results have important implications for future mitigation of nitrate in this region. The manuscript is overall well written, and I only have several small comments.

Reply: We would like to thank the reviewer for the insightful comments, which help us in improving the quality of our work. Please find the responses to individual comments below.

1. The measurements at the urban and suburban sites were conducted in different years? Did the author compare the meteorological differences between 2018 and 2019? Are there any influences on your conclusions?

Reply: We thank the reviewer for the valuable suggestion. The measurements at the urban and suburban sites were conducted in 2018 and 2019, respectively. Generally, the meteorological factors have important influences on the nitrate pollution. We have compared the average values of wind speed (WS), relative humidity (RH) and temperature ($T$) in the sampling periods at the urban site and suburban site as shown in **Table S6** in the revised manuscript as follows. The average wind speeds at the urban and suburban sites were generally below 2 m s$^{-1}$, thus, we mainly focus on the local production which simulated by the box model. The RH, $T$ and photolysis frequency were set as the observation data in the simulation, which represented the actual meteorological condition. The simulated results also demonstrate the influence of meteorological condition, and showed no influence on our conclusions.

**Table S6. The concentrations of chemical components (average ± standard deviation) and meteorological parameters during the investigated periods at the GIG and Heshan sites**

| Site | GIG | Heshan |
|---|---|---|
| $PM_1$ ($\mu g\ m^{-3}$) | 41.7±23.1 | 40.6 ±15.5 |
| Organic ($\mu g\ m^{-3}$) | 16.9±9.0 | 21.6 ± 9.0 |
| $SO_4^{2-}$ ($\mu g\ m^{-3}$) | 10.1±4.6 | 6.9 ± 1.8 |
| $NO_3^-$ ($\mu g\ m^{-3}$) | 6.1±5.8 | 3.9 ± 3.0 |
| $NH_4^+$ ($\mu g\ m^{-3}$) | 5.0±3.0 | 3.5 ± 1.5 |
| $Cl^-$ ($\mu g\ m^{-3}$) | 0.6±0.54 | 0.8 ± 1.3 |
| BC ($\mu g\ m^{-3}$) | 3.2±1.1 | 4.0 ± 1.6 |
| WS (m/s) | 1.9±0.9 | 1.6±0.7 |
| RH (%) | 76.2±14.9 | 59.5±14.3 |
| $T$ (°C) | 23.0±2.6 | 23.2±3.2 |

2. The urban site is approximately 80 km from the suburban site. Could the authors provide the wind rose plots during the two years to see if there is transport between the two sites. Or the authors can compare the total PM concentrations in the same year to see if the episodes occurred during the same period. This will also affect the conclusion in this study.

Reply: We agree with your comment that regional transport is also important for nitrate pollution. We have compared the wind rose plots at the urban and suburban sites as shown in the following Figure R5.

The wind direction at the GIG site was mainly from the north, and the wind speed was frequently lower than 4 m s$^{-1}$. The wind direction at the Heshan site was mainly from north and northwest, and the wind speed was lower than 4 m s$^{-1}$. As GIG site is in the northeast of Heshan site, the transport between the two sites was weak from the results of wind rose plots.

[Figure]

Figure R5. The wind rose plot at the urban (GIG) site and suburban (Heshan) site in the study periods.

3. "ammonia" in Figure 2 should be "ammonium", same in Figure 3.

Reply: We have changed the legend "ammonia" to **"ammonium"** in **Figure 2 and Figure 3** in the revised manuscript as follows.

[Figure]

**Figure 2. Temporal variations of the mass concentration of the major chemical components in PM₁ including nitrate (NO₃⁻), sulfate (SO₄²⁻), ammonium (NH₄⁺), black carbon (BC), chloride (Cl⁻) and organics at (a) GIG site and (b) Heshan site. The black dashed rectangle represents the investigated period which had complete set of data.**

[Figure]

**Figure 3. The mass concentration ratio of NO₃⁻/SO₄²⁻ (top) and fractions of major chemical components (bottom) in PM₁ at (a) GIG site and (b) Heshan site.**